# Forward-Backward Reasoning in Large Language Models for Mathematical Verification

## Abstract

Chain-of-Thought (CoT) prompting in large language models (LLMs) has shown promising performance on mathematical reasoning tasks. Recently, Self-Consistency (Wang et al., 2023) samples a diverse set of reasoning chains with different answers and chooses the answer by majority voting. Though effective, its performance cannot be further improved by sampling more reasoning chains. To address this problem, we propose to integrate backward reasoning into answer verification. We first mask a number in the question by $\mathbf{x}$. The LLM is then asked to predict the masked number with a candidate answer $\hat{A}_c$ embedded in the template: "*If we know the answer to the above question is $\{\hat{A}_c\}$, what is the value of unknown variable $\mathbf{x}$?*" The LLM is expected to predict the masked number successfully if the provided candidate answer is correct. To further improve performance, we propose FOBAR (FOrward-BAckward Reasoning) to combine forward and backward reasoning for verifying candidate answers. Experiments are performed on six standard mathematical data sets and three LLMs (*text-davinci-003*, *GPT-3.5-Turbo*, *GPT-4*). Results show that FOBAR achieves state-of-the-art performance. In particular, FOBAR outperforms Self-Consistency which uses forward reasoning alone, demonstrating that combining forward and forward reasoning is better. It also outperforms existing verification methods, verifying the effectiveness of using the simple template in backward reasoning and the proposed combination.

## 1 Introduction

*Few-shot prompting* (or *in-context learning*) (Brown et al., 2020; Min et al., 2022; Chen et al., 2022) allows pre-trained large language models (LLMs) (Chowdhery et al., 2022; OpenAI, 2023; Wu et al., 2023) to generalize well to unseen tasks. This is performed by concatenating a few examples (e.g., question-answer pairs) as a prompt, and then appending the testing question. Compared with traditional methods such as finetuning (Howard & Ruder, 2018; Devlin et al., 2019), few-shot prompting is more desirable as the large LLM (e.g., 175 billion parameters in *GPT-3*) does not need to be re-trained. However, it is still challenging for LLMs to generate answers to mathematical questions by simply prompting the question-answer pairs, as mathematics is more complex and often many steps are required to derive the answer.

Recently, Wei et al. (2022) propose *chain-of-thought (CoT) prompting* for LLMs, which generates explicit intermediate steps that are required to reach the final answer. Specifically, each in-context example is augmented with several thinking steps described in natural language. A few examples are concatenated as a CoT prompt. In inference, the testing question is appended to the prompt and fed to an LLM. The LLM is expected to imitate the in-context examples, i.e., generate several reasoning steps before giving the answer. CoT Prompting has achieved promising performance on mathematical reasoning tasks (Wei et al., 2022; Wang et al., 2023; Zheng et al., 2023; Zhang et al., 2023b). Recently, many works have been proposed to improve its effectiveness (Fu et al., 2023; Zheng et al., 2023; Zhou et al., 2023; Yao et al., 2023; Pitis et al., 2023) and efficiency (Zhang et al., 2023b; Kojima et al., 2022; Diao et al., 2023; Lu et al., 2022).

Self-Consistency Wang et al. (2023) is a simple but effective approach to improve CoT prompting. Using temperature sampling (Ackley et al., 1985; Ficler & Goldberg, 2017), it samples a diverse set of reasoning chains which may lead to multiple candidate answers. The one that receives the most votes is then selected as the final answer. Figure 1 shows the testing accuracy of Self-Consistency

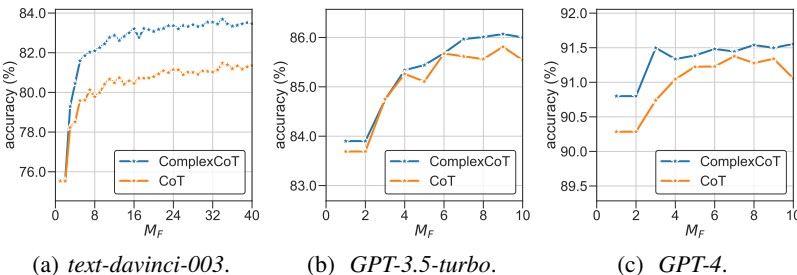

(a) *text-davinci-003*.  (b) *GPT-3.5-turbo*.  (c) *GPT-4*.

**Figure 1:** Testing accuracy of Self-Consistency versus number of sampling paths ($M_F$) averaged on six data sets.

with different numbers ($M_F$) of sampling paths, averaged over six data sets and three LLMs (the experimental setup is in Section 4.1). As can be seen, while Self-Consistency is effective, simply sampling more reasoning paths may not lead to performance improvement, particularly when $M_F$ is large.

In this paper, we introduce *backward reasoning* (or *backward chaining*) (Pettit & Sugden, 1989; Russell & Norvig, 1995; Khot et al., 2021; Liang et al., 2021; Yu et al., 2023) for verifying the candidate answers. While Self-Consistency uses *forward reasoning* for verification (i.e., starting with a question, the LLM generates multiple reasoning steps to reach its answer), backward reasoning works backward from a candidate answer to the antecedent for checking if any data supports this answer.

To use backward reasoning for verifying answers, we mask a specific number in the question and ask the LLM to predict the masked number when a candidate answer $\hat{A}_c$ is provided. Intuitively, the correct candidate answer can predict the masked number more accurate than the incorrect answers (Figure 5 in Section 4.5). Specifically, we mask a number in the question by "**x**" and append a template "*If we know the answer to the above question is* $\{\hat{A}_c\}$*, what is the value of unknown variable* **x**?" to form a backward question. This is then fed to the LLM to generate multiple steps before predicting the value of **x**. As the ground-truth value of **x** is available, we can check correctness of the prediction. If the prediction matches the ground-truth of **x**, the candidate answer is likely to be correct. Unlike Self-Verification (Weng et al., 2022) which needs the assistance of an LLM to rewrite the question to a declarative statement (e.g., "How many hours does he spend on TV and reading in 4 weeks?" with a candidate answer of 36 is rewritten to "He spends 36 hours on TV and reading in 4 weeks"), we append a simple template to the question without rewriting.

Backward reasoning and forward reasoning are complementary. We propose FOrward-BAckward Reasoning (FOBAR) to combine them (Figure 2). In the *forward* direction, we estimate the probability $\mathbb{P}(\hat{A}_c; \text{forward})$ of a candidate answer by the proportion of votes it gets. In the *backward* direction, for each candidate answer $\hat{A}_c$, we create several questions for backward reasoning by masking numbers and sample a set of backward reasoning chains to predict the masked number. The vote of $\hat{A}_c$ is the number of chains that predict the masked number correctly. We estimate the probability $\mathbb{P}(\hat{A}_c; \text{backward})$ as the proportion of votes $\hat{A}_c$ gets in the backward direction. By combining backward and forward reasoning, we estimate the probability $\mathbb{P}(\hat{A}_c)$ as the geometric mean of forward and backward probabilities. Extensive experiments on six data sets and three OpenAI's LLMs (*text-davinci-003* (OpenAI, 2022a), *GPT-3.5-Turbo* (OpenAI, 2022a), *GPT-4* (OpenAI, 2023)) show that FOBAR achieves state-of-the-art performance.

Our contributions can be summarized as follows. (i) We introduce backward reasoning to mathematical verification, where a simple template is proposed to create backward questions by masking numbers in the original question when a candidate answer is provided. We further design a CoT prompt for the LLM to predict the masked number and estimate the probability of the candidate answer based on the number of correct chains in the backward direction. (ii) We propose FOBAR which combines forward and backward reasoning for verifying candidate answers. (iii) Experimental results on six standard mathematical benchmarks and three LLMs show that FOBAR achieves SOTA performance. In particular, FOBAR outperforms Self-Consistency which uses forward reasoning alone, demonstrating that combining forward and backward reasoning together is better. Additionally, FOBAR outperforms Self-Verification, confirming that using the simple template in backward reasoning and the proposed combination is more effective.

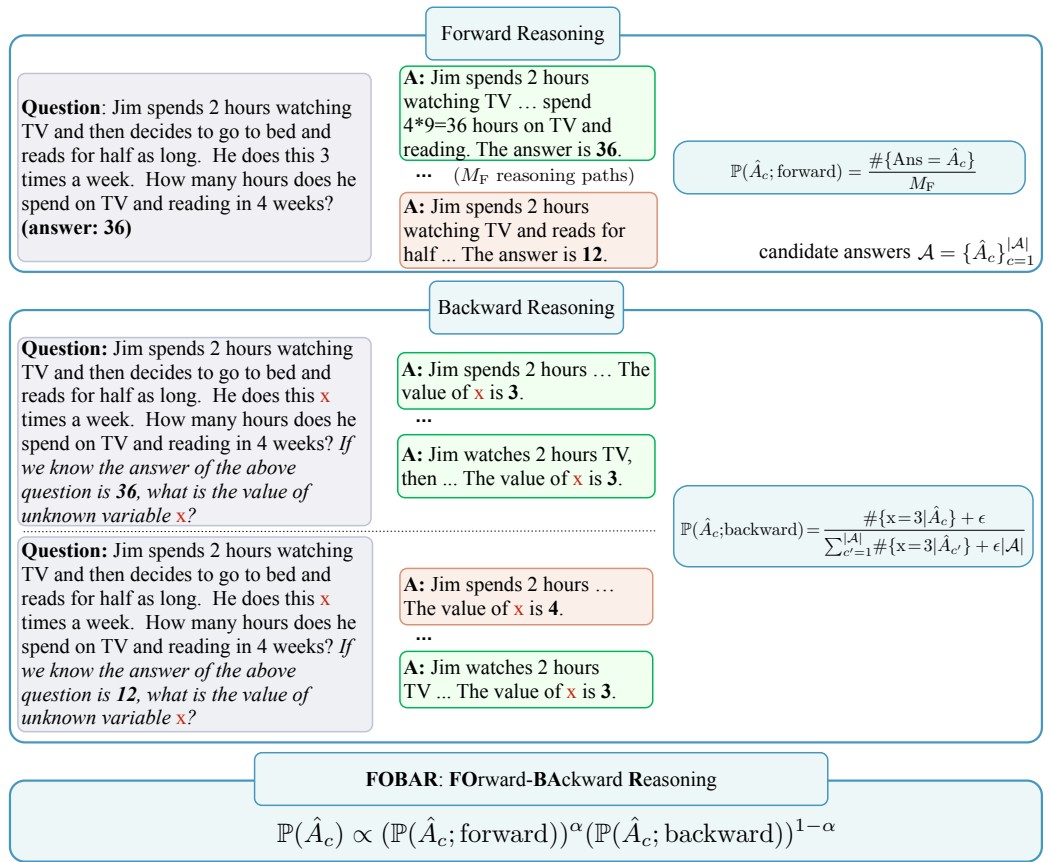

**Figure 2:** Overview of forward reasoning, backward reasoning, and the proposed FOBAR. The detailed procedure is shown in Algorithm 1.

## 2 RELATED WORK

**Few-Shot Prompting** (or **In-Context Learning (ICL)**). Brown et al. (2020), Min et al. (2022), Chen et al. (2022), Liu et al. (2022), Rubin et al. (2022), Liu et al. (2023) use large language models (LLMs) to solve a task by feeding $K$ examples as part of input. The $K$ examples are concatenated as a prompt

$$\mathbf{P}_{\text{ICL}} = \text{"Question: } Q^{(1)} \text{ \textbackslash n Answer: } A^{\star(1)} \ldots \text{ Question: } Q^{(K)} \text{ \textbackslash n Answer: } A^{\star(K)}\text{"},$$

where $Q^{(i)}$ and $A^{\star(i)}$ are the question and answer, respectively. In inference, a new question $Q$ is appended to the prompt as "$\mathbf{P}_{\text{ICL}}$ \n Question: $Q$ \n Answer:" and fed to the LLM for generating output sequences. An answer extractor is used to extract the prediction $\hat{A}$ from the output (e.g., the number after the last "Answer:" (Brown et al., 2020)). Prompting is more efficient than model finetuning (Howard & Ruder, 2018; Devlin et al., 2019) in computation and memory, as the LLM is fixed and shared across tasks. This can be crucial as LLMs are usually very large (e.g., *GPT-3* (Brown et al., 2020) has 175 billion parameters). Few-shot prompting has demonstrated promising performance on a variety of tasks (Brown et al., 2020; Rubin et al., 2022; Liu et al., 2022; Ye et al., 2023; Xu et al., 2023a). However, for mathematical tasks (e.g., *GSM8K* (Cobbe et al., 2021))) which are complex as many steps are required to reach the answer, concatenating question-answer pairs as a prompt is still challenging for LLMs to generate the answer directly.

**Chain-of-Thought (CoT) Prompting**. Wei et al. (2022) proposes to augment question-answer pairs with intermediate steps such that the LLM can solve questions step-by-step. Specifically, each in-context example is a triplet $(Q^{(i)}, R^{(i)}, A^{\star(i)})$, where $R^{(i)}$ is a natural language description of the steps leading from $Q^{(i)}$ to $A^{\star(i)}$. In inference, a new question $Q$ is appended to the prompt as:

$$\mathbf{P}_{\text{CoT}} = \text{"Question: } Q^{(1)} \text{ \textbackslash n Answer: } R^{(1)}, A^{\star(1)} \ldots \text{ Question: } Q^{(K)} \text{\textbackslash n Answer: } R^{(K)}, A^{\star(K)}\text{"} \quad (1)$$

and "$\mathbf{P}_{\text{CoT}}$ \n Question: $Q$ \n Answer:" is fed to the LLM for generating both the reasoning chain $R$ and answer $A$. CoT prompting has achieved state-of-the-art performance in a wide variety of tasks

(Wei et al., 2022; Kojima et al., 2022; Fu et al., 2023; Zhang et al., 2023b; Wang et al., 2023; Zheng et al., 2023; Zhou et al., 2023; Diao et al., 2023; Zhang et al., 2023c). Recently, many methods have been proposed to reduce the required expert knowledge in designing reasoning chains. For example, Zero-shot CoT (Kojima et al., 2022) uses the magic prompt "*Let's think step by step*" for answering questions, which is further combined with few-shot prompting in AutoCoT (Zhang et al., 2023b).

Recently, many works (Fu et al., 2023; Zheng et al., 2023; Madaan et al., 2023; Paul et al., 2023; Shinn et al., 2023; Welleck et al., 2023; Zhou et al., 2023; Chen et al., 2023; Zhang et al., 2023a) have been proposed to improve the quality of reasoning chains in CoT prompting. Complex CoT (Fu et al., 2023) selects examples with more steps as in-context examples, while PHP (Zheng et al., 2023) iteratively uses the previous answers as hints in prompting. These methods can be viewed as *forward reasoning*, which starts from the question and generates a reasoning chain to reach the answer. Instead of taking a single reasoning chain by greedy decoding, Self-Consistency (Wang et al., 2023) samples a diverse set of chains and obtains a set of candidate answers. The final answer is then selected by voting.

**Backward Reasoning** (or **Backward Chaining**) (Pettit & Sugden, 1989; Russell & Norvig, 1995; Khot et al., 2021; Liang et al., 2021; Yu et al., 2023) starts with an answer and work backward to determine the sequence of steps or conditions necessary to reach this answer. Backward reasoning is particularly useful in domains when the answer is known, e.g., in automated theorem provers (Russell & Norvig, 1995; Rocktäschel & Riedel, 2016; Wang & Deng, 2020; Kazemi et al., 2023; Poesia & Goodman, 2023). Here, we use backward reasoning to verify the candidate answer by checking whether a masked number can be successfully predicted when the candidate answer is provided. Recently, Self-Verification (Weng et al., 2022) also uses backward reasoning to verify answers. It first rewrites the question with an answer to a declarative statement and then asks the LLM to predict the masked number. RCoT (Xue et al., 2023) regenerates the question conditioning on the answer and detects whether there is factual inconsistency in the constructed question. Compared with Self-Verification and RCoT, we simply append a template to the original question without additional rewriting and reconstruction. Note that RCoT needs to reconstruct a sequence of tokens in the question, which is challenging to check the correctness and three complex steps are required. This complicated checking method also leads to inaccurate verification. In contrast, the proposed FOBAR just needs to predict the masked number and check whether the number is predicted correctly by string comparison, which is much simpler and more accurate. Furthermore, FOBAR combines forward and backward reasoning together for verification, while RCoT uses backward alone.

## 3  FORWARD-BACKWARD REASONING FOR MATHEMATICAL VERIFICATION

In this section, we first generate a set of candidate answers in the forward direction by temperature sampling (Ackley et al., 1985; Ficler & Goldberg, 2017) and estimate each answer's probability based on the votes it receives (Section 3.1). Next, we create questions for backward reasoning and ask the LLM to predict the masked number (Section 3.2). Finally, we propose FOBAR (Section 3.3), which combines forward and backward reasoning for verifying the candidate answers. Figure 2 provides an overview of FOBAR. The detailed procedure is shown in Algorithm 1.

### 3.1  FORWARD REASONING

*Forward reasoning* starts with a question and generates multiple intermediate steps toward the answer. Specifically, for a question $Q$, we prepend it with a base prompt $\mathbf{P}_\mathrm{F}$ (e.g., Chain-of-Thought (CoT) prompting (Wei et al., 2022) or ComplexCoT prompting (Fu et al., 2023)) and feed the tuple $(\mathbf{P}_\mathrm{F}, Q)$ to the LLM for generating a reasoning chain and candidate answer. Using temperature sampling, we sample $M_\mathrm{F}$ candidate reasoning chains $\{R_i\}_{i=1}^{M_\mathrm{F}}$ and extract the corresponding candidate answers $\{A_i\}_{i=1}^{M_\mathrm{F}}$ (Figure 2(top)). Let $\mathcal{A} = \{\hat{A}_c\}_{c=1}^{|\mathcal{A}|}$ be the set of answers deduplicated from $\{A_i\}_{i=1}^{M_\mathrm{F}}$. Unlike greedy decoding (Wei et al., 2022), we may have several different candidate answers (i.e., $|\mathcal{A}| > 1$). We propose to estimate the probability that candidate answer $\hat{A}_c \in \mathcal{A}$ is correct by the proportion of votes it receives from the various reasoning paths:

$$\mathbb{P}_{\text{forward}}(\hat{A}_c) = \frac{1}{M_\mathrm{F}} \sum_{i=1}^{M_\mathrm{F}} \mathbb{I}(A_i = \hat{A}_c), \tag{2}$$

where $\mathbb{I}(\cdot)$ is the indicator function. Choosing the $\hat{A}_c$ with the largest $\mathbb{P}_{\text{forward}}(\hat{A}_c)$ recovers the state-of-the-art method of Self-Consistency (Wang et al., 2023). In Section 3.3, we propose to refine the estimated probabilities by combining forward and backward reasoning. Furthermore, as can be seen from Figure 1, the performance of Self-Consistency saturates when $M_{\text{F}}$ is sufficiently large. Thus, simply sampling more reasoning paths brings negligible performance improvements.

## 3.2 BACKWARD REASONING

In backward reasoning, we mask a number contained in the question and ask the LLM to predict the masked number with a provided candidate answer. Specifically, suppose that question $Q$ involves $N_Q$ numbers $\{\text{num}^{(n)}\}_{n=1}^{N_Q}$. We replace each of them one by one with **x**. The resultant masked question $\hat{Q}^{(n)}$ is then concatenated with the following template, which contains a candidate answer $\hat{A}_c \in \mathcal{A}$:

> $\mathcal{T}(\hat{A}_c)$ = *If we know the answer to the above question is $\{\hat{A}_c\}$, what is the value of unknown variable* x?

Each $(\hat{Q}^{(n)}, \mathcal{T}(\hat{A}_c))$ pair is called a *backward question*. In total, we obtain $N_Q$ backward questions. An example is shown in the following. Note that Self-Verification (Weng et al., 2022) needs the assistance of an LLM to rewrite a (question, answer) pair to a declarative statement.[1] Here, the use of a template is simpler and avoids possible mistakes (an example involving rewriting mistake by Self-Verification is shown in Appendix A).

---

**Example 3.1: Backward questions.**

**Question**: Jim spends x hours watching TV and then decides to go to bed and reads for half as long. He does this 3 times a week. How many hours does he spend on TV and reading in 4 weeks?
*If we know the answer to the above question is $\{\hat{A}_c\}$, what is the value of unknown variable* x?
**Question**: Jim spends 2 hours watching TV and then decides to go to bed and reads for half as long. He does this x times a week. How many hours does he spend on TV and reading in 4 weeks?
*If we know the answer to the above question is $\{\hat{A}_c\}$, what is the value of unknown variable* x?
**Question**: Jim spends 2 hours watching TV and then decides to go to bed and reads for half as long. He does this 3 times a week. How many hours does he spend on TV and reading in x weeks?
*If we know the answer to the above question is $\{\hat{A}_c\}$, what is the value of unknown variable* x?

---

To predict the masked number, we prepend the backward question with a prompt $\mathbf{P}_{\text{B}}$, which consists of several (backward) question-answer demos with reasoning chains. An example question-answer demo is shown in Example G.1 of Appendix G.

For $n = 1, \ldots, N_Q$, we feed $(\mathbf{P}_{\text{B}}, \hat{Q}^{(n)}, \mathcal{T}(\hat{A}_c))$ to the LLM, which then imitates the in-context examples in $\mathbf{P}_{\text{B}}$ and generates a reasoning chain for the prediction of the masked number. We sample $M_B$ such reasoning chains, with predictions $\{\widehat{\text{num}}_{c,b}^{(n)}\}_{b=1}^{M_{\text{B}}}$. For each candidate answer $\hat{A}_c$, we count the number of times that the masked number is correctly predicted: $Z_c = \sum_{n=1}^{N_Q} \sum_{b=1}^{M_{\text{B}}} \mathbb{I}(\widehat{\text{num}}_{c,b}^{(n)} = \text{num}^{(n)})$. The probability that candidate answer $\hat{A}_c$ is correct is estimated as

$$\mathbb{P}_{\text{backward}}(\hat{A}_c) = \frac{Z_c + \epsilon}{\sum_{c'=1}^{|\mathcal{A}|} Z_{c'} + \epsilon|\mathcal{A}|}, \tag{3}$$

where $\epsilon = 10^{-8}$ is a small positive constant (to avoid division by zero). One can simply choose the $\hat{A}_c$ with the largest $\mathbb{P}_{\text{backward}}(\hat{A}_c)$ as prediction. A more effective method, as will be shown in Section 3.3, is to combine the probabilities obtained from both forward and backward reasoning.

## 3.3 FOBAR (**FO**RWARD AND **BA**CKWARD **R**EASONING)

As forward and backward reasoning are complementary, we propose to combine them for verification. Intuitively, a candidate answer is likely to be correct when it receives many votes in forward reasoning and also helps the LLM in predicting the masked numbers correctly in backward reasoning. We estimate the probability that $\hat{A}_c$ is correct as

$$\mathbb{P}(\hat{A}_c) \propto \left(\mathbb{P}_{\text{forward}}(\hat{A}_c)\right)^{\alpha} \left(\mathbb{P}_{\text{backward}}(\hat{A}_c)\right)^{1-\alpha}, \tag{4}$$

---

[1]For example, "How many hours does he spend on TV and reading in 4 weeks?" with the candidate answer of 36 is rewritten to "He spends 36 hours on TV and reading in 4 weeks".

where the weight $\alpha \in [0, 1]$. When $\alpha = 1$, it reduces to Self-Consistency (Wang et al., 2023); when $\alpha = 0$, it reduces to backward reasoning. In the experiments, we combine the two forward and backward probabilities by the geometric mean (i.e., $\alpha = 0.5$). Finally, we select the prediction as $\arg\max_{\hat{A}_c \in \mathcal{A}} \mathbb{P}(\hat{A}_c)$. The whole procedure is shown in Algorithm 1.

---

**Algorithm 1** FOBAR (**FO**rward and **BA**ckward **R**easoning).

---

**Require:** number of reasoning chains sampled $M_F$ and $M_B$, prompts $\mathbf{P}_F$ and $\mathbf{P}_B$; $\epsilon = 10^{-8}$; smoothing factor $\alpha = 0.5$;
1: given a testing question $Q$ involving $N_Q$ numbers;
2: feed $(\mathbf{P}_F, Q)$ to LLM, sample $M_F$ reasoning chains with candidate answers $\{A_i\}_{i=1}^{M_F}$;
3: deduplicate $\{A_i\}_{i=1}^{M_F}$ to $\mathcal{A} = \{\hat{A}_c\}_{c=1}^{|\mathcal{A}|}$;
   *Forward Reasoning:*
4: compute probability $\mathbb{P}_{\text{forward}}(\hat{A}_c) = \frac{1}{M_F} \sum_{i=1}^{M_F} \mathbb{I}(A_i = \hat{A}_c)$ for each $\hat{A}_c \in \mathcal{A}$;
   *Backward Reasoning:*
5: **for** each $\hat{A}_c \in \mathcal{A}$ **do**
6:     **for** $n = 1, \ldots, N_Q$ **do**
7:         create $\hat{Q}^{(n)}$ by masking the $n$th number $\text{num}^{(n)}$ in $Q$;
8:         feed $(\mathbf{P}_B, \hat{Q}^{(n)}, \mathcal{T}(\hat{A}_c))$ to LLM;
9:         sample $M_B$ predictions $\{\widehat{\text{num}}_{c,b}^{(n)}\}_{b=1}^{M_B}$;
10:    **end for**
11:    count number of correct predictions: $Z_c = \sum_{n=1}^{N_Q} \sum_{b=1}^{M_B} \mathbb{I}(\widehat{\text{num}}_{c,b}^{(n)} = \text{num}^{(n)})$;
12: **end for**
13: compute probability $\mathbb{P}_{\text{backward}}(\hat{A}_c) = \frac{Z_c + \epsilon}{\sum_{c'=1}^{|\mathcal{A}|} Z_{c'} + \epsilon|\mathcal{A}|}$ for each $\hat{A}_c \in \mathcal{A}$;
14: compute probability $\mathbb{P}(\hat{A}_c) \propto (\mathbb{P}_{\text{forward}}(\hat{A}_c))^{\alpha} (\mathbb{P}_{\text{backward}}(\hat{A}_c))^{1-\alpha}$ for each $\hat{A}_c \in \mathcal{A}$;
15: **return** $\arg\max_{\hat{A}_c \in \mathcal{A}} \mathbb{P}(\hat{A}_c)$.

---

## 4 EXPERIMENTS

### 4.1 SETUP

**Datasets.** Experiments are performed on six standard mathematical data sets (Table 3 in Appendix B): (i) *AddSub* (Hosseini et al., 2014), (ii) *MultiArith* (Roy & Roth, 2015), (iii) *SingleEQ* (Koncel-Kedziorski et al., 2015), (iv) *SVAMP* (Patel et al., 2021), (v) *GSM8K* (Cobbe et al., 2021), (vi) *AQuA* (Ling et al., 2017). The first three are from the Math World Problem Repository (Koncel-Kedziorski et al., 2016), while the last three are proposed more recently. Questions in *AddSub* and *SingleEQ* are easier and do not need multi-step calculations, while *MultiArith*, *SVAMP*, *GSM8K*, and *AQuA* are more challenging as many steps are required.

**Baselines.** We compare the proposed FOBAR with (i) In-Context Learning (ICL) using question-answer pairs as demos (Brown et al., 2020), and recent chain-of-thought (CoT) prompting methods, including: (ii) CoT prompting (Wei et al., 2022); (iii) ComplexCoT prompting (Fu et al., 2023) which selects demonstrations with complex reasoning steps; (iv) RE2 (Xu et al., 2023b) which re-reads the question in the prompt. (v) PHP (Zheng et al., 2023) which iteratively uses the previous answers as hints in designing prompts. (vi) RCoT (Xue et al., 2023) which reconstructs the question based on the candidate answer and checks the factual inconsistency for verification. (vii) Self-Consistency (Wang et al., 2023), which samples multiple reasoning chains and selects the answer by majority voting; (viii) Self-Verification (Weng et al., 2022), which chooses the top-2 candidate answers obtained from Self-Consistency and re-ranks them based on the verification scores;

**Implementation Details.** We experiments with three LLMs: (i) *text-davinci-003* (OpenAI, 2022a), (ii) *GPT-3.5-Turbo* (OpenAI, 2022b), and (iii) *GPT-4* (OpenAI, 2023). *GPT-3.5-Turbo* and *GPT-4* are more powerful than *text-davinci-003*. In both forward and backward reasoning, the temperature for sampling is set to 0.7 as in Wang et al. (2023). The $\alpha$ value in (4) is set to 0.5. For *text-davinci-003*, $M_F$ is set to 40 as in (Wang et al., 2023; Zheng et al., 2023); whereas the more powerful LLMs (*GPT-3.5-Turbo* and *GPT-4*) use a smaller $M_F$ value of 10 (as can be seen from Figure 1). $M_B$ is set to 8 for all three LLMs. The proposed method is general and can be integrated into any

**Table 1:** Testing accuracies on six data sets using three LLMs. For each LLM, methods are grouped according to the base prompt they used, where the best in each group is in **bold**. Results with [†] are from the original publications. "–" means the result is not reported in the original publications.

| | | AddSub | MultiArith | SingleEQ | SVAMP | GSM8K | AQuA | Avg |
|---|---|---|---|---|---|---|---|---|
| *text-davinci-003* | ICL (Brown et al., 2020) | 90.4 | 37.6 | 84.3 | 69.1 | 16.9 | 29.1 | 54.5 |
| | *(CoT Prompting)* | | | | | | | |
| | CoT (Wei et al., 2022) | 91.4 | 93.6 | 92.7 | 79.5 | 55.8 | 46.5 | 76.6 |
| | PHP[†] (Zheng et al., 2023) | 91.1 | 94.0 | 93.5 | 81.3 | 57.5 | 44.4 | 77.0 |
| | RE2[†] (Xu et al., 2023b) | 91.7 | 93.3 | 93.3 | 81.0 | 61.6 | 44.5 | 77.6 |
| | Self-Consistency (Wang et al., 2023) | 91.7 | 95.9 | 94.5 | 83.1 | 67.9 | **55.1** | 81.4 |
| | Self-Verification (Weng et al., 2022) | 87.4 | 95.3 | 92.9 | 82.2 | 59.8 | 37.4 | 75.8 |
| | FOBAR | **91.9** | **100.0** | **96.1** | **86.8** | **70.8** | **55.1** | **83.5** |
| | *(ComplexCoT Prompting)* | | | | | | | |
| | ComplexCoT (Fu et al., 2023) | 88.9 | 95.3 | 93.7 | 78.0 | 67.7 | 48.8 | 78.7 |
| | PHP[†] (Zheng et al., 2023) | **91.6** | 96.6 | 95.0 | 83.7 | 68.4 | 53.1 | 81.4 |
| | Self-Consistency (Wang et al., 2023) | 89.4 | 98.5 | 91.1 | 82.7 | **79.1** | **58.7** | 83.2 |
| | Self-Verification (Weng et al., 2022) | 89.9 | 95.5 | 94.1 | 80.1 | 72.0 | 38.2 | 78.3 |
| | FOBAR | 90.6 | **100.0** | **95.3** | **87.0** | 78.7 | **58.7** | **85.0** |
| *GPT-3.5-Turbo* | ICL (Brown et al., 2020) | 88.6 | 87.6 | 88.8 | 80.6 | 32.2 | 31.1 | 68.2 |
| | *(CoT Prompting)* | | | | | | | |
| | CoT (Wei et al., 2022) | 89.4 | 97.9 | 92.9 | 84.2 | 77.2 | 54.3 | 82.7 |
| | RE2[†] (Xu et al., 2023b) | 89.9 | 96.5 | **95.3** | 80.0 | 80.6 | 58.3 | 83.4 |
| | Self-Consistency (Wang et al., 2023) | **90.6** | 98.6 | 93.1 | 86.4 | 81.9 | **62.6** | 85.5 |
| | Self-Verification (Weng et al., 2022) | 90.4 | 97.4 | 92.9 | 83.1 | 74.9 | 60.6 | 83.2 |
| | FOBAR | 89.4 | **99.3** | 94.5 | **88.9** | **85.1** | **62.6** | **86.6** |
| | *(ComplexCoT Prompting)* | | | | | | | |
| | Complex CoT (Fu et al., 2023) | 87.9 | 98.3 | **94.5** | 81.1 | 80.7 | 59.1 | 83.6 |
| | RCoT[†] (Xue et al., 2023) | 88.2 | – | 93.0 | 84.9 | 84.6 | 53.3 | – |
| | PHP[†] (Zheng et al., 2023) | 85.3 | 98.0 | 92.9 | 83.1 | 85.1 | 60.6 | 84.2 |
| | Self-Consistency (Wang et al., 2023) | 88.1 | 98.8 | **94.5** | 85.0 | 86.4 | 63.0 | 86.0 |
| | Self-Verification (Weng et al., 2022) | 87.9 | 96.6 | 93.3 | 81.0 | 78.2 | 61.4 | 83.1 |
| | FOBAR | **88.4** | **99.8** | 94.3 | **88.5** | **87.4** | **63.4** | **87.0** |
| *GPT-4* | ICL (Brown et al., 2020) | 92.1 | 98.6 | 94.3 | 90.9 | 48.5 | 48.0 | 78.7 |
| | *(CoT Prompting)* | | | | | | | |
| | CoT (Wei et al., 2022) | **92.7** | **99.0** | 95.7 | 92.9 | 93.4 | 69.7 | 90.6 |
| | Self-Consistency (Wang et al., 2023) | 92.2 | **99.0** | 95.9 | 93.3 | 94.8 | **71.3** | 91.1 |
| | Self-Verification (Weng et al., 2022) | **92.7** | **99.0** | 95.7 | 93.1 | 93.7 | 70.1 | 90.7 |
| | FOBAR | 92.4 | **99.0** | 96.1 | **94.1** | **95.4** | **71.3** | **91.4** |
| | *(ComplexCoT Prompting)* | | | | | | | |
| | Complex CoT (Fu et al., 2023) | **91.9** | 98.3 | 94.5 | 92.4 | 95.1 | 72.4 | 90.8 |
| | PHP[†] (Zheng et al., 2023) | 89.6 | 98.1 | 93.1 | 91.9 | 95.5 | **79.9** | 91.3 |
| | Self-Consistency (Wang et al., 2023) | 91.4 | 98.5 | **94.7** | 93.4 | 96.2 | 75.2 | 91.6 |
| | Self-Verification (Weng et al., 2022) | 91.6 | 98.5 | **94.7** | 93.0 | 95.7 | 75.6 | 91.5 |
| | FOBAR | **91.9** | **98.6** | **94.7** | **94.4** | **96.4** | 75.2 | **91.9** |

prompting method. Here, we choose the standard CoT prompting (Wei et al., 2022) and ComplexCoT prompting (Fu et al., 2023) as the base prompt.

## 4.2 RESULTS

Table 1 shows the testing accuracies. As can be seen, for all three LLMs, FOBAR with ComplexCoT prompting achieves the highest average accuracy. When using CoT as the base prompt, FOBAR always has higher average accuracy than Self-Consistency, demonstrating the effectiveness of integrating backward reasoning into verification. Furthermore, FOBAR outperforms Self-Verification almost all the time, except on *AddSub* (with CoT as base prompt) and *AQuA* (with ComplexCoT) on *GPT-4*, demonstrating that using a simple template in backward reasoning and the proposed combination is better. Note also that methods based on CoT are better than ICL by a large margin, confirming the effectiveness of CoT. FOBAR (with either CoT or ComplexCoT) on *GPT-4* achieves

the highest testing accuracy (averaged over six data sets). Moreover, for all three LLMs, FOBAR using ComplexCoT as the base prompt achieves higher accuracy than using CoT on average.

### 4.3 USEFULNESS OF FORWARD AND BACKWARD REASONING

In this section, we perform an ablation study on forward (FO) and backward (BA) reasoning. We consider the four combinations: (i) using neither forward nor backward reasoning (i.e., greedy decoding (Wei et al., 2022)); (ii) use only forward reasoning (i.e., Self-Consistency); (iii) use only backward reasoning; (iv) use both forward and backward reasoning (i.e., the proposed FOBAR). Table 2 shows the testing accuracies averaged over the six data sets for three LLMs. As can be seen, in all settings, using forward or backward reasoning is consistently better than using neither of them. Moreover, combining both forward and backward reasoning is always the best.

**Table 2:** Average testing accuracies with different combinations of forward and backward reasoning.

| | FO | BA | t.d.003 | GPT-3.5 | GPT-4 |
|---|---|---|---|---|---|
| CoT | ✗ | ✗ | 76.6 | 82.7 | 90.6 |
| | ✓ | ✗ | 81.4 | 85.5 | 91.1 |
| | ✗ | ✓ | 82.1 | 86.2 | 91.2 |
| | ✓ | ✓ | **83.5** | **86.6** | **91.4** |
| ComplexCoT | ✗ | ✗ | 78.7 | 83.6 | 90.8 |
| | ✓ | ✗ | 83.2 | 86.0 | 91.6 |
| | ✗ | ✓ | 81.3 | 86.3 | 91.8 |
| | ✓ | ✓ | **85.0** | **87.0** | **91.9** |

### 4.4 VARIATION WITH $\alpha$

In this experiment, we study how the weight $\alpha$ in (4) affects performance. Figure 3 shows the testing accuracies (averaged over the six data sets) w.r.t. $\alpha \in [0, 1]$ using the three LLMs. As can be seen, FOBAR is insensitive to $\alpha$ over a large range for all three LLMs. Hence, in the experiment, we use $\alpha = 0.5$, corresponding to the geometric mean of the forward and backward probabilities.

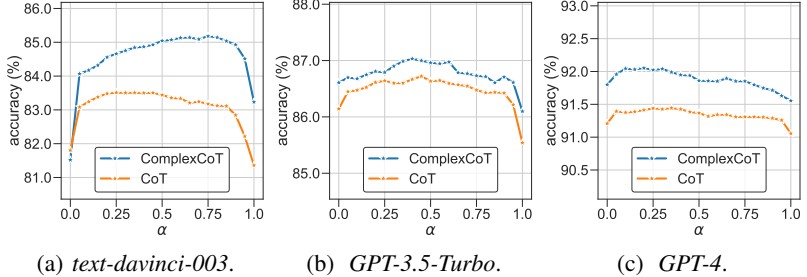

(a) *text-davinci-003*.  (b) *GPT-3.5-Turbo*.  (c) *GPT-4*.
**Figure 3:** Testing accuracy (averaged over the six data sets) of FOBAR with $\alpha$.

Alternatively, one can also combine the forward and backward probabilities by the arithmetic mean, i.e., $\mathbb{P}(\hat{A}_c) = \frac{1}{2}\big(\mathbb{P}_{\text{forward}}(\hat{A}_c) + \mathbb{P}_{\text{backward}}(\hat{A}_c)\big)$. Figure 4 shows the testing accuracies obtained (averaged over six data sets) for the three LLMs. As can be seen, the arithmetic mean achieves comparable performance to the geometric mean. Hence, Figures 3 and 4 together suggest that FOBAR is robust to the combination of forward and backward probabilities.

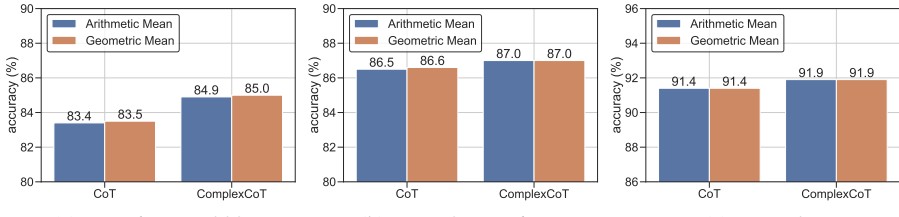

(a) *text-davinci-003*.  (b) *GPT-3.5-Turbo*.  (c) *GPT-4*.
**Figure 4:** Testing accuracy of FOBAR (averaged over the six data sets) with geometric/arithmetic mean of forward and backward probabilities.

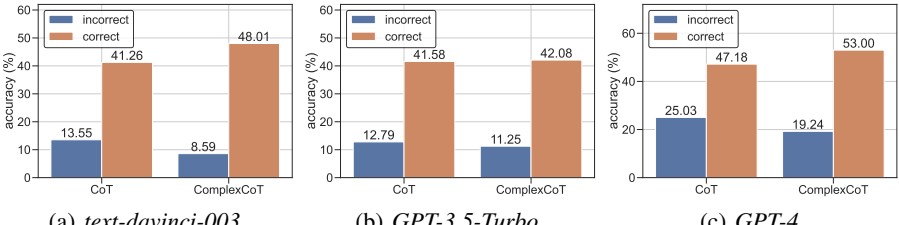

(a) *text-davinci-003*.  (b) *GPT-3.5-Turbo*.  (c) *GPT-4*.
**Figure 5:** Accuracy (averaged over all backward questions across the six data sets) of predicting the masked number in backward questions with incorrect/correct candidate answers.

### 4.5 ACCURACY IN PREDICTING THE MASKED NUMBER

In this section, we study whether the correct candidate answer can predict the masked number more accurately than the incorrect candidate answers in answering backward questions. Figure 5 shows the accuracy of predicting the masked number in backward questions given incorrect/correct candidate answers. As can be seen, the correct candidate answer can predict the masked number much more accurately than the incorrect candidate answers.

### 4.6 VARIATION WITH $M_F$

In this experiment, we study how the performance of FOBAR varies with the number of reasoning chains $M_F$ sampled in forward reasoning. Figure 6 shows the testing accuracies (averaged over the six data sets) for the three LLMs. As can be seen, using a very small $M_F$ (e.g., $\leq 5$) is clearly undesirable, but the performance quickly saturates. This suggests one can use a small $M_F$ for reducing computational cost. Moreover, note that the accuracy curves of FOBAR are higher than those of Self-Consistency in Figure 1, again demonstrating the effectiveness of integrating backward reasoning into verification.

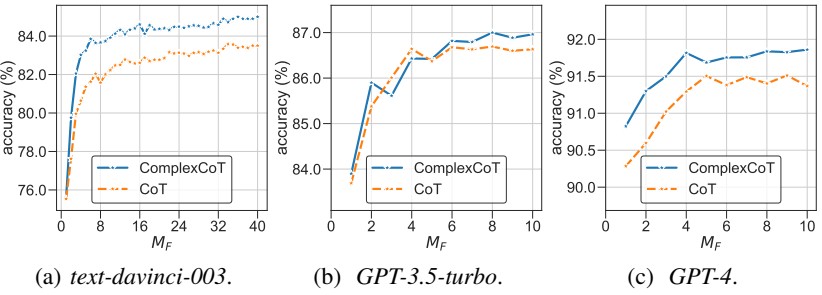

(a) *text-davinci-003*.   (b) *GPT-3.5-turbo*.   (c) *GPT-4*.

**Figure 6:** Testing accuracy of FOBAR (averaged over the six data sets) with $M_F$.

### 4.7 VARIATION WITH $M_B$

In this experiment, we study how the performance of FOBAR varies with the number of reasoning chains $M_B$ sampled in backward reasoning. Figure 7 shows the testing accuracies (averaged over the six data sets) for the three LLMs. Note that $M_B = 0$ corresponds to using only forward reasoning. As can be seen, using a very small $M_B$ (e.g., $\leq 4$) is clearly undesirable, but the performance quickly saturates. Hence, using a small $M_B$ can achieve good performance and efficiency.

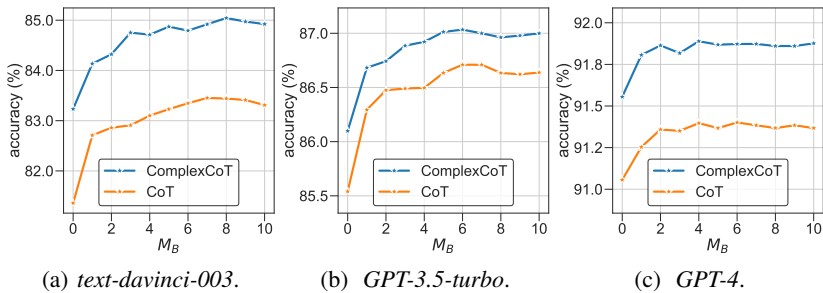

(a) *text-davinci-003*.   (b) *GPT-3.5-turbo*.   (c) *GPT-4*.

**Figure 7:** Testing accuracy of FOBAR (averaged over the six data sets) with $M_B$.

## 5 CONCLUSION

In this paper, we study the problem of verifying answers for mathematical tasks. We introduce backward reasoning into verification, where a simple template is introduced to create questions and a prompt is designed to ask the LLM to predict a masked number when a candidate answer is provided. Furthermore, we proposed FOBAR to combine forward and backward reasoning for verification. Extensive experiments are performed on six standard data sets and three LLMs. Results show that the proposed FOBAR achieves state-of-the-art performance. In particular, FOBAR outperforms Self-Consistency (Wang et al., 2023), which uses forward reasoning alone, demonstrating that combining forward and backward reasoning is more effective in verification. FOBAR also outperforms Self-Verification (Weng et al., 2022), verifying the effectiveness of using the proposed simple template in backward reasoning and the proposed combination.

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

## A  EXAMPLE INVOLVING MISTAKE IN REWRITING QUESTIONS FOR SELF-VERIFICATION (WENG ET AL., 2022)

> **Question:** A class of 50 students has various hobbies. 10 like to bake, 5 like to play basketball, and the rest like to either play video games or play music. How many like to play video games if the number that like to play music is twice the number that prefer playing basketball? (answer: 25)

We mask the first number (i.e., 50) by x and a candidate answer 25 is provided. As below, we show the backward question obtained by Self-Verification and FOBAR. We can see that Self-Verification makes a mistake in re-writing the question into a declarative statement, while a simple template in FOBAR does not need extra rewriting.

> **Question (Self-Verification):** A class of x students has various hobbies. 10 like to bake, 5 like to play basketball, and the rest like to either play video games or play music. The number of people who like to play video games is equal to the number of people who prefer playing basketball multiplied by two. The number of people who like to play video games is 25. What is the answer of x?
>
> **Question (FOBAR):** A class of x students has various hobbies. 10 like to bake, 5 like to play basketball, and the rest like to either play video games or play music. How many like to play video games if the number that like to play music is twice the number that prefer playing basketball? *If we know the answer to the above question is 25, what is the value of unknown variable x?*

## B  DATA SETS

Table 3 shows a summary of data sets used in the experiments.

## C  EXTENSION TO OTHER TYPES OF TASKS

Extending the proposed backward verification and FOBAR to non-arithmetic tasks is possible and follows a similar line. Similar to that for mathematical tasks, we mask informative word/token/character and then ask the LLM to predict. We give two examples (Date Understanding reasoning task (Wei et al., 2022; Srivastava & et al. , 400+ authors) and Last-Letter-Concatenation reasoning task (Wei et al., 2022)) as follows.

### Example 1: Date Understanding

Question: Yesterday was April 30, 2021. What is the date today in MM/DD/YYYY?

**Table 3:** Data sets used in the experiments.

| | #samples | $N_Q$ (mean $\pm$ std) | example |
|---|---|---|---|
| *AddSub* | 395 | $2.6 \pm 0.7$ | Benny picked 2 apples and Dan picked 9 apples from the apple tree. How many apples were picked in total? |
| *MultiArith* | 600 | $3.1 \pm 0.3$ | Katie picked 3 tulips and 9 roses to make flower bouquets. If she only used 10 of the flowers though, how many extra flowers did Katie pick? |
| *SingleEQ* | 508 | $2.2 \pm 0.7$ | Joan went to 4 football games this year. She went to 9 football games last year. How many football games did Joan go to in all? |
| *SVAMP* | 1000 | $2.8 \pm 0.7$ | Rachel has 4 apple trees. She picked 7 apples from each of her trees. Now the trees have a total 29 apples still on them. How many apples did Rachel pick in all? |
| *GSM8K* | 1319 | $3.8 \pm 1.6$ | A robe takes 2 bolts of blue fiber and half that much white fiber. How many bolts in total does it take? |
| *AQuA* | 254 | $2.9 \pm 1.3$ | If the population of a city increases by 5% annually, what will be the population of the city in 2 years time if its current population is 78000? Answer Choices: (A) 81900 (B) 85995 (C) 85800 (D) 90000 (E) None of these |

Candidate Answer: 05/01/2021 (correct), 05/02/2021 (wrong)

Backward question for the candidate answer 05/01/2021: Yesterday was April 30, 2021. What is the date **x** in MM/DD/YYYY? If we know the answer to the above question is **05/01/2021**, what is the English word at **x**?

Backward question for the candidate answer 05/02/2021: Yesterday was April 30, 2021. What is the date **x** in MM/DD/YYYY? If we know the answer to the above question is **05/02/2021**, what is the English word at **x**?

The LLM is more likely to predict the word "today" given the correct candidate answer 05/01/2021. We leave the details as future work.

**Example 2: Last-Letter-Concatenation**

Question: Take the last letters of each word in "Whitney Erika Tj Benito" and concatenate them.

Candidate Answer: yajo (correct), yaji (wrong)

Backward question for candidate answer yajo: Take the last letters of each word in "Whitney Erika Tj Benit[]" and concatenate them. If we know the answer to the above question is **yajo**, what is the character at []?

Backward question for candidate answer yaji: Take the last letters of each word in "Whitney Erika Tj Benit[]" and concatenate them. If we know the answer to the above question is **yaji**, what is the character at []?

We use "[]" to mask the character instead of "x" (which is also a character). The LLM is more likely to predict "o" correctly at [] given the candidate answer yajo than yaji. Hence, the proposed FOBAR can be used in other types of tasks by masking the informative word/token/character in the questions.

Note that FOBAR is a novel method to combine forward and backward reasoning for verification (i.e., $\mathbb{P}(\hat{A}_c) \propto (\mathbb{P}_{\text{forward}}(\hat{A}_c))^\alpha (\mathbb{P}_{\text{backward}}(\hat{A}_c))^{1-\alpha}$). The proposed method is general and can be integrated into existing verification methods (such as RCoT (Xue et al., 2023) and Self-Verification (Weng et al., 2022)) for non-arithmetic reasoning tasks.

# D  ADDITIONAL EXPERIMENTS

## D.1  COMPARISON BETWEEN FOBAR AND TRAINING A VERIFIER

Compared with Cobbe et al. (2021), which trains an LLM for verifying answers, FOBAR has two advantages. (i) **(training-free)** Compared with training an LLM for verifying candidate answers, which is computationally expensive and labor-intensive in collecting extra annotation data, backward reasoning for verifying is training-free and requires no additional data collection. (ii) **(more effective)** As training the GPT-3 (175B) model is extremely expensive and their code is not publicly available, we compare our FOBAR with the result reported in Figure 5 of (Cobbe et al., 2021), where the

candidate answers are generated by GPT-3. Table 4 shows the accuracy on GSM8K. As shown, FOBAR consistently performs much better than the trained verifier (+14.8).

**Table 4:** Comparison between FOBAR and a trained verifier on *GSM8K*.

| | |
|---|---|
| Training GPT-3 (175B) for Verification (Cobbe et al., 2021) | 56.0 |
| FOBAR (text-davinci-003 + CoT) | 70.8 |
| FOBAR (text-davinci-003 + ComplexCoT) | 78.7 |
| FOBAR (GPT-3.5-Turbo + CoT) | 85.1 |
| FOBAR (GPT-3.5-Turbo + ComplexCoT) | 87.4 |
| FOBAR (GPT-4 + CoT) | 95.4 |
| FOBAR (GPT-4 + ComplexCoT) | 96.4 |

## D.2 ADDITIONAL EXPERIMENT WITH DIFFERENT SEEDS

We conducted an additional experiment on GSM8K using GPT-3.5-Turbo with ComplexCoT prompting. We repeat the experiment with three different seeds. Table 5 shows the testing accuracy. As can be seen, FOBAR performs better than Self-Consistency on all three seeds. Furthermore, the improvement of FOBAR over Self-Consistency is statistically significant (according to the pairwise t-test, with a p-value of 0.0013).

**Table 5:** Accuracy of Self-Consistency and FOBAR on GSM8K with three different seeds.

| | seed 1 | seed 2 | seed 3 | mean ($\pm$std) |
|---|---|---|---|---|
| Self-Consistency | 86.4 | 86.7 | 86.2 | $86.43 \pm 0.25$ |
| FOBAR | 87.4 | 87.6 | 87.1 | $87.37 \pm 0.25$ |

## D.3 COMPARISON BETWEEN FOBAR AND STEP-BY-STEP VERIFICATION

Recent works (Lightman et al., 2023; Ling et al., 2023) propose verifying the steps of forward reasoning chains. Lightman et al. (2023) propose to label exclusively steps of forward reasoning chains generated by LLMs. The labeled data are then used to train an LLM for verification. Compared with Lightman et al. (2023), which is computationally expensive in training an LLM and labor-intensive in labeling data, our backward reasoning is training-free for verification and requires no additional data annotation.

Ling et al. (2023) propose a natural language-based deductive reasoning format that allows the LLM to verify **forward** reasoning steps. Different from (Ling et al., 2023), we use **backward** reasoning to verify the candidate answers instead of the steps in forward chains. As backward and forward reasoning are complementary, our backward verification can be combined with their step-by-step forward verification. We replace the forward verification in our FOBAR (Eq. (4)) with step-by-step forward verification proposed by Ling et al. (2023), and conducted additional experiments on AddSub, GSM8K, and AQuA using GPT-3.5-Turbo. Table 6 shows the testing accuracy. As can be seen, combining backward verification consistently boosts performance.

**Table 6:** Accuracy of FOBAR which combining forward verification + backward verification.

| | AddSub | GSM8K | AQuA |
|---|---|---|---|
| Self-Consistency | 88.1 | 86.4 | 63.0 |
| Self-Consistency + Backward Verification | **88.4** | **87.4** | **63.4** |
| NP (Ling et al., 2023) | 93.67 | 87.05 | 70.34 |
| NP + Backward Verification | **93.92** | **87.89** | **71.65** |
| NP + Deductive Verification + UPV (Ling et al., 2023) | 93.54 | 86.01 | 69.49 |
| NP + Deductive Verification + UPV + Backward Verification | **93.92** | **87.19** | **70.86** |

# E CASE STUDY

## E.1 CASES THAT SELF-CONSISTENCY FAILS BUT FOBAR SUCCEEDS

We conducted a qualitative analysis on SingleEq using text-davinci-003 with CoT prompting. We observe that a common characteristic of problems that Self-Consistency fails but FOBAR succeeds is: Problems are difficult to solve in a forward direction, but the correctness of a candidate answer can be easily verified in a backward manner. An example is shown below.

Question: The sum of three consecutive odd numbers is 69. What is the smallest of the three numbers?

Ground-truth answer: 21

Candidate answers generated by Self-Consistency: 21 (16 times), 23 (24 times)

An example of wrong forward reasoning chains: The sum of three consecutive odd numbers is 69. We can use the formula n + (n+2) + (n+4) = 69 to solve for n. We get n = 23. This is the smallest of the three numbers. The answer is 23.

Question for backward verification: The sum of three consecutive odd numbers is $x$. What is the smallest of the three numbers? If we know the answer to the above question is 21, what is the value of unknown variable $x$?

For the above backward question, it is easy for the LLM to calculate the value of $x$. We sample 10 backward reasoning chains to solve the question, and all of them predict $x = 69$. Hence, in this case, backward verification is simpler than solving the original question.

## E.2 ANALYSIS ON CASES THAT SELF-CONSISTENCY FAILS

We conducted an additional analysis on Self-Consistency using the GPT-3.5-Turbo with ComplexCoT prompting. Table 7 shows the number of failure problems in Self-Consistency, and the number of failure problems which have no correct chains or at least one correct chain. We can see that, in total, about 60% of failure problems have correct chains in Self-Consistency, while about 40% of problems have no correct ones and thus cannot be solved by backward verification.

Table 7: Analysis on the cases that Self-Consistency fails.

|  | AddSub | MultiArith | SingleEQ | SVAMP | GSM8K | AQuA | Total |
|---|---|---|---|---|---|---|---|
| #fails | 47 | 7 | 28 | 150 | 179 | 94 | 505 |
| #has no correct answers | 28 | 0 | 14 | 57 | 60 | 52 | 211 |
| #has the correct answer | 19 | 7 | 14 | 93 | 119 | 42 | 294 |

## E.3 HARD CASES IN BACKWARD REASONING

We analyzed the questions in GSM8K (using GPT-3.5-Turbo with ComplexCoT ) that Self-Consistency succeeds but FOBAR fails. There are 24 such questions. We give an example below, where backward reasoning fails to predict the masked numbers.

Question: Manolo bought 5 lollipops and 4 candies that cost $3.20. If each lollipop costs $0.40, how much will 10 lollipops and 10 candies cost him?

Ground-truth answer: 7

Candidate answers generated by Self-Consistency: 12 (once), 16 (4 times), 7 (5 times)

Number of times that the masked number is correctly predicted in backward reasoning : 12 (6 times), 16 (0 times), 7 (0 times)

# F LIMITATIONS AND FUTURE WORKS

In this paper, we focus on mathematical tasks, which often require a number of reasoning steps to reach the answer and are thus challenging. The proposed template (i.e., masking a number in

the question and then asking the LLM to predict the masked number) is simple, general, and can be applied out-of-the-box to any mathematical task. For mathematical tasks, we choose to mask numbers as they are informative; However, for non-arithmetic tasks, how to select information words/marks/characters still needs to be manually designed, as two examples are given in Appendix C. Extending FOBAR to non-arithmetic tasks is future work.

## G   QUESTION-ANSWER DEMO OF BACKWARD REASONING

---

**Example G.1: Example question-answer demo with reasoning chain.**

**Question:** Randy has 60 mango trees on his farm. He also has x less than half as many coconut trees as mango trees. How many trees does Randy have in all on his farm? *If we know the answer to the above question is 85, what is the value of unknown variable* x*?*
**A:** Let's think step by step. We know that Randy has 60 mango trees on his farm. We also know that he has x less than half as many coconut trees as mango trees. Let's use C to represent the number of coconut trees. So we can write: C = (1/2)*60 - x = 30 - x. The total number of trees on Randy's farm is the sum of the number of mango trees and coconut trees: 60 + (30 - x) = 90 - x. We are given that the total number of trees on Randy's farm is 85, so we can write: 90 - x = 85. Solving for x, we get: x = 5. The value of x is 5.

---

