# OpenReview forum: "Forward-Backward Reasoning in Large Language Models for Mathematical Verification"
_ICLR.cc/2024/Conference — Submitted to ICLR 2024_

### Official Review · Reviewer_nUu4 · 2023-10-29

**Soundness:** 4 excellent
**Presentation:** 4 excellent
**Contribution:** 3 good
**Rating:** 8
**Confidence:** 4

**Summary:**

This paper proposes a reasoning verification method for language models for mathematical reasoning tasks. The method, FOBAR, combines forward (self-consistency) and backward chaining for verification. The main idea is to ask the model to go from a candidate answer to a masked quantity in the question. The authors experiment with 3 OpenAI models (text-davinci-003, gpt-3.5-turbo, and gpt-4), finding positive results in 6 mathematical reasoning datasets (including GSM8k). Moreover, an ablation shows that the method is complementary to self-consistency, with the best results coming from a combination of both.

**Strengths:**

The paper is well-written, well-motivated and addresses a current topic - unsupervised verification of chain-of-thought reasoning. The idea is sound for the domain it is proposed for, of mathematical reasoning. Many current mathematical reasoning datasets are amenable to this, as shown by the extensive evaluation.

The experimental results are quite strong - notably, this almost sets a new state-of-the-art on GSM8k using GPT-4, for instance.

Also, this idea is likely to "age well", i.e. it gets better and more relevant as LLMs become more capable. This is a noticeable feature in the current phase of AIs, where many papers have relevance for at most a few months. In contrast, the idea of backward verification using the LLM itself can potentially be applied to more challenging datasets of the future, as LLMs themselves become more capable of both solving problems and also of producing coherent verification traces.

**Weaknesses:**

The main weakness in my opinion is the relatively narrow scope of the method (mathematical reasoning, and even then mostly on more numerical tasks). While I can see the idea being applicable to other domains, it's not obvious how to do so (if there were other compelling examples, e.g. in logical reasoning tasks, I believe the authors would have likely shown some of them). This is one disadvantage compared to Self-Consistency which, while less effective for math as shown here, is very widely applicable.

For results, I think the paper currently misses a qualitative discussion on what failure modes FOBAR addresses. While the idea is intuitive, it's not obvious where exactly self-consistency fails as a verification method, when FOBAR would be expected to succeed.

One baseline that is missing is the recent paper from Ling et al, 2023 on "Deductive Verification of Chain-of-Thought Reasoning". Their method is a forward verification method, and it would be interesting to see if (1) the gains from FOBAR + Self-Consistency alone are higher than what they get, and (2) if their method could even be a better drop-in replacement for the "forward" part in FOBAR. If there are directly comparable numbers on their paper, I'd strongly suggest citing them here. If cost is an issue, even doing this just for AquA, where you have the lowest absolute results and thus most room for improvement, could be already interesting.

**Questions:**

A minor point that I'd suggest improving on is the mathematical notation. This is not a barrier to understanding, since the idea is very simple, but will just help standardize papers in the community. For each candidate answer, there is an associated "correct" random variable. Equations 2 and 3 describe estimators for this unknown quantity. The estimator's name is what you'd put a hat on, not the variable. I'd indicate forward/backward as subscripts in the estimator's name, rather than as a "parameter".

Questions:
- Are there any examples where self-consistency fails but FOBAR succeeds that help give insights into the failure modes that FOBAR addresses?
- When self-consistency fails, is it usually when the model consistently proposes a wrong answer, or proposes a completely divergent set of answers that are then ultimately sampled at random?
- Do you believe FOBAR can be applied outside of numerical mathematical reasoning tasks? If so, what other tasks could be the closest potential targets?
- Have you observed cases where backward verification is especially hard, while forward verification is not?
- Do you have a sense of how often backward verification discards correct solutions? In other words, of the remaining failures, how many are cases where the base model fails to give any correct answer at all, versus cases where one of the answers is correct but ends up discarded by FOBAR?
 * Related to this last question, one suggestion that should be simple to implement is computing an "oracle verifier", which takes any correct answer from the model if there is one, and otherwise returns any answer. This might help show the potential to improve results by improving the verifier (as FOBAR/self-consistency try to do), versus cases where we would only see improvements if the base model were to improve.

---

> ### Author Response · Authors · 2023-11-18
> **Reply to Reviewer nUu4 (1/4)**
>
> Thank you for your positive review and valuable comments. We address your concerns as follows.
>
> > Q1. "The main weakness in my opinion is the relatively narrow scope of the method (mathematical reasoning, and even then mostly on more numerical tasks). While I can see the idea being applicable to other domains, it's not obvious how to do so (if there were other compelling examples, e.g. in logical reasoning tasks, I believe the authors would have likely shown some of them). This is one disadvantage compared to Self-Consistency which, while less effective for math as shown here, is very widely applicable."
> >
> > "Do you believe FOBAR can be applied outside of numerical mathematical reasoning tasks? If so, what other tasks could be the closest potential targets?"
>
> A1. (i) In this paper, **our focus is on mathematical tasks**, which often require a number of reasoning steps to reach the answer and are thus challenging. The proposed backward verification is **simple, general, and can be applied out-of-the-box to any mathematical task.**
>
> (ii) **Extension to non-arithmetic tasks is possible** and follows a similar line. Similar to that for mathematical tasks, we **mask informative word/token/character** and then ask the LLM to predict. We give **two examples (Date Understanding reasoning task [1, 2] and Last-Letter-Concatenation reasoning task [1])** as follows.
>
> `Example 1: Date Understanding`
>
> *Question*: Yesterday was April 30, 2021. What is the date today in MM/DD/YYYY?
>
> *Candidate Answer*: 05/01/2021 (correct), 05/02/2021 (wrong)
>
> *Backward question for the candidate answer 05/01/2021*:
> Yesterday was April 30, 2021. What is the date **x** in MM/DD/YYYY? If we know the answer to the above question is **05/01/2021**, what is the English word at **x**?
>
> *Backward question for the candidate answer 05/02/2021*:
> Yesterday was April 30, 2021. What is the date **x** in MM/DD/YYYY? If we know the answer to the above question is **05/02/2021**, what is the English word at **x**?
>
> The LLM is more likely to predict the word "today" given the correct candidate answer 05/01/2021. We leave the details as future work.
>
> `Example 2: Last-Letter-Concatenation`
>
> *Question*: Take the last letters of each word in "Whitney Erika Tj Benito" and concatenate them.
>
> *Candidate Answer*: yajo (correct), yaji (wrong)
>
> *Backward question for candidate answer yajo*:
> Take the last letters of each word in "Whitney Erika Tj Benit[]" and concatenate them. If we know the answer to the above question is **yajo**, what is the character at []?
>
> *Backward question for candidate answer yaji*:
> Take the last letters of each word in "Whitney Erika Tj Benit[]" and concatenate them. If we know the answer to the above question is **yaji**, what is the character at []?
>
> Note that we use "[]" to mask the character instead of "x" (which is also a character). The LLM is more likely to predict "o" correctly at [] given the candidate answer yajo than yaji. Hence, the proposed FOBAR can be used in other types of tasks by masking the informative word/token/character in the questions.
>
> (iii) Note that one contribution of the proposed FOBAR is a **novel** method to combine forward and backward reasoning for verification (i.e., $\mathbb{P}(\hat{A}_c) \propto (\mathbb{P}\_\text{forward}(\hat{A}_c) )^\alpha (\mathbb{P}\_\text{backward}(\hat{A}_c) )^{1-\alpha}$). The proposed method is **general** and **can be integrated into existing verification methods** (such as RCoT [3] and Self-Verification [4]) for non-arithmetic reasoning tasks.

---

> ### Author Response · Authors · 2023-11-18
> **Reply to Reviewer nUu4 (2/4)**
>
> > Q2. "For results, I think the paper currently misses a qualitative discussion on what failure modes FOBAR addresses. While the idea is intuitive, it's not obvious where exactly self-consistency fails as a verification method, when FOBAR would be expected to succeed."
> >
> > "Are there any examples where self-consistency fails but FOBAR succeeds that help give insights into the failure modes that FOBAR addresses?"
>
> A2. As suggested, we conducted a qualitative analysis on SingleEq using text-davinci-003 with CoT prompting. We observe that a common characteristic of problems that  Self-Consistency fails but FOBAR succeeds is: **Problems are difficult to solve in a forward direction,
> but the correctness of a candidate answer can be easily verified in a backward manner**. An example is shown below.
>
> *Question*: The sum of three consecutive odd numbers is 69. What is the smallest of the three numbers?
>
> *Ground-truth answer*: 21
>
> *Candidate answers generated by Self-Consistency*: 21 (16 times), 23 (24 times)
>
> *An example of wrong forward reasoning chains*: The sum of three consecutive odd numbers is 69. We can use the formula n + (n+2) + (n+4) = 69 to solve for n. We get n = 23. This is the smallest of the three numbers. The answer is 23.
>
> *Question for backward verification*: The sum of three consecutive odd numbers is ${\bf x}$. What is the smallest of the three numbers? If we know the answer to the above question is $21$, what is the value of unknown variable ${\bf x}$?
>
> For the above backward question, it is easy for the LLM to calculate the value of **x**. We sample 10 backward reasoning chains to solve the question, and all of them predict ${\bf x}=69$. Hence, in this case, **backward verification is simpler than solving the original question.**
>
> ---
>
> > Q3. "One baseline that is missing is the recent paper from Ling et al, 2023 on "Deductive Verification of Chain-of-Thought Reasoning". Their method is a forward verification method, and it would be interesting to see if (1) the gains from FOBAR + Self-Consistency alone are higher than what they get, and (2) if their method could even be a better drop-in replacement for the "forward" part in FOBAR. If there are directly comparable numbers on their paper, I'd strongly suggest citing them here. If cost is an issue, even doing this just for AquA, where you have the lowest absolute results and thus most room for improvement, could be already interesting."
>
> **A3.** Thanks for bringing this recent paper [5] to our attention. As suggested, we replace the forward part in our FOBAR with the forward verification proposed in [5], and conducted additional experiments on AddSub, GSM8K, and AQuA using the GPT-3.5-Turbo. Table below shows the testing accuracy. As can be seen, **combining our backward verification with their forward verifications consistently boosts the performance.**
>
> \begin{array}{l c c c}
> \hline
> &  \text{AddSub} & \text{GSM8K} & \text{AQuA} \newline
> \hline
> \text{Self-Consistency}  & 88.1 & 86.4 & 63.0 \newline
>  \text{Self-Consistency + Backward Verification} &\textbf{ 88.4} & \textbf{87.4} & \textbf{63.4}\newline
> \hline
> \text{NP [1]}  & 93.67 &  87.05 & 70.34 \newline
> \text{NP [1] + Backward Verification}  & \textbf{93.92} & \textbf{87.89} & \textbf{71.65} \newline
> \hline
> \text{NP + DV + UPV [1]} & 93.54 & 86.01 & 69.49  \newline
> \text{NP + DV + UPV [1] + Backward Verification}& \textbf{93.92} &  \textbf{87.19} & \textbf{70.86} \newline
> \hline
> \end{array}

---

> ### Author Response · Authors · 2023-11-18
> **Reply to Reviewer nUu4 (3/4)**
>
> > Q4. "A minor point that I'd suggest improving on is the mathematical notation. This is not a barrier to understanding, since the idea is very simple, but will just help standardize papers in the community. For each candidate answer, there is an associated "correct" random variable. Equations 2 and 3 describe estimators for this unknown quantity. The estimator's name is what you'd put a hat on, not the variable. I'd indicate forward/backward as subscripts in the estimator's name, rather than as a "parameter"."
>
> **A4.** Thanks for your suggestions, and we revised the paper accordingly.
>
> ---
>
> > Q5. "When self-consistency fails, is it usually when the model consistently proposes a wrong answer, or proposes a completely divergent set of answers that are then ultimately sampled at random?"
>
> **A5.** We conducted an additional analysis on Self-Consistency using the GPT-3.5-Turbo with ComplexCoT prompting. Table below shows the number of failure problems in Self-Consistency, and the number of failure problems with no correct chains or at least one correct chain. We can see that, in total, about 60\% of the failure problems have correct chains in Self-Consistency, while about 40\% of the problems have no correct ones and thus cannot be solved by backward verification.
>
> \begin{array}{cccccccc}
> \hline
> & \text{AddSub} & \text{MultiArith} & \text{SingleEQ} & \text{SVAMP} & \text{GSM8K} & \text{AQuA} & \text{Total}\newline
> \hline
> \text{\\#fails} & 47 & 7 & 28 & 150 & 179 & 94 & 505\newline
> \hline
> \text{\\#has no correct answers} & 28 & 0 & 14 & 57 & 60 & 52 & 211 \newline
> \text{\\#has the correct answer} & 19 & 7 & 14 & 93 & 119 & 42 & 294 \newline
> \hline
> \end{array}
>
> ---
>
> > Q6. "Have you observed cases where backward verification is especially hard, while forward verification is not?"
>
> **A6.** As suggested, we analyzed the questions in GSM8K (using GPT-3.5-Turbo with ComplexCoT) that  Self-Consistency succeeds but FOBAR fails. There are 24 such questions. We give an example below, where backward reasoning fails to predict the masked numbers.
>
> *Question*: Manolo bought 5 lollipops and 4 candies that cost \\$3.20. If each lollipop costs \\$0.40, how much will 10 lollipops and 10 candies cost him?
>
> *Ground-truth answer*: 7
>
> Candidate answers generated by Self-Consistency:  12 (once), 16 (4 times), 7 (5 times)
>
> Number of times that the masked number is correctly predicted in backward reasoning: 12 (6 times), 16 (0 times), 7 (0 times)
>
> ---
>
> > Q7. "Do you have a sense of how often backward verification discards correct solutions? In other words, of the remaining failures, how many are cases where the base model fails to give any correct answer at all, versus cases where one of the answers is correct but ends up discarded by FOBAR?"
>
> **A7.** As suggested, we did an analysis on the failure cases of FOBAR using the GPT-3.5-Turbo with ComplexCoT prompting to study how often FOBAR discards correct solutions. Table below shows the results. As shown, in total, there are 400 problems  FOBAR fails. Among them, nearly half of them have no correct candidate answers, which cannot be improved by enhancing backward verification. Furthermore, there are 189 questions that have forward chains leading to the correct answer but FOBAR fails.
>
> \begin{array}{cccccccc}
> \hline
> & \text{AddSub} & \text{MultiArith} & \text{SingleEQ} & \text{SVAMP} & \text{GSM8K} & \text{AQuA} & \text{Total} \newline
> \hline
> \text{\\#samples} & 395 & 580 & 508 & 1000 & 1319 & 254 & 4056\newline
> \hline
> \text{\\#succeeds} & 349 & 579 & 479 & 885 & 1153 & 161 & 3606\newline
> \text{\\#fails} & 46 & 1 & 29 & 115 & 116 & 93 & 400 \newline
> \hline
> \text{\\#has no correct answers} & 28 & 0 & 14 & 57 & 60 & 52 & 211  \newline
> \text{\\#has the correct answer} & 18 & 1 & 15 & 58 & 56 &  41 & 189 \newline
> \hline
> \end{array}

---

> ### Author Response · Authors · 2023-11-18
> **Reply to Reviewer nUu4 (4/4)**
>
> > Q8. "Related to this last question, one suggestion that should be simple to implement is computing an "oracle verifier", which takes any correct answer from the model if there is one, and otherwise returns any answer. This might help show the potential to improve results by improving the verifier (as FOBAR/self-consistency try to do), versus cases where we would only see improvements if the base model were to improve."
>
> **A8.** We conducted an additional analysis to show the potential of improving performance by enhancing the verifier. Table below shows the accuracy of Oracle Verifier (i.e., pick the ground-truth answer if it is a candidate answer generated by Self-Consistency). We can see that there is a large gap between Oracle Verifier and FOBAR/Self-Consistency. Hence, **improving the verifier is a potential method to boost performance.**
>
> \begin{array}{c|c |c c c c c c c}
> \hline
>  & \text{method} & \text{AddSub} & \text{MultiArith} & \text{SingleEQ} & \text{SVAMP} & \text{GSM8K} & \text{AQuA} & \text{Avg} \newline
> \hline
>  & \text{Self-Consistency} & 91.7 &   95.9 & 94.5 & 83.1 &67.9& 55.1& 81.4 \newline
> \text{text-davinci-003+CoT}&\text{FOBAR}   &91.9 & 100.0 & 96.1& 86.8 & 70.8 & 55.1 & 83.5 \newline
> & \text{Oracle Verifier} & 97.0 & 100.0  & 98.0 & 92.8 & 84.6 & 85.8 & 93.0 \newline
> \hline
>   & \text{Self-Consistency} & 89.4 &98.5  & 91.1& 82.7 & 79.1 & 58.7 & 83.2 \newline
> \text{text-davinci-003+ComplexCoT} & \text{FOBAR}   & 90.6 &100.0& 95.3 &87.0 & 78.7 & 58.7 & 85.0 \newline
> & \text{Oracle Verifier} & 95.4 & 100.0 & 97.4 & 94.5 & 92.2 & 82.7 & 93.7 \newline
> \hline
>  & \text{Self-Consistency} &90.6 & 98.6 & 93.1 & 86.4 & 81.9 & 62.6 & 85.5\newline
> \text{GPT-3.5-turbo+CoT} & \text{FOBAR} & 89.4 & 99.3 & 94.5 & 88.9 & 85.1 & 62.6 & 86.6 \newline
> & \text{Oracle Verifier} & 93.9 & 100.0 & 96.5 & 93.9 & 92.2 & 89.8 & 94.4\newline
> \hline
>  & \text{Self-Consistency} & 88.1 &  98.8 & 94.5 & 85.0&86.4 & 63.0 &  86.0 \newline
> \text{GPT-3.5-turbo+ComplexCoT} &\text{FOBAR} & 88.4 & 99.8& 94.3 & 88.5 & 87.4 & 63.4 & 87.0 \newline
> & \text{Oracle Verifier} & 92.9 & 100.0 & 97.2 & 94.3 & 95.5 & 79.5 & 93.2 \newline
> \hline
>  & \text{Self-Consistency} & 92.2 &  99.0 & 95.9 & 93.3 & 94.8 & 71.3 & 91.1\newline
> \text{GPT-4+CoT}& \text{FOBAR} & 92.4 & 99.0 & 96.1& 94.1 & 95.4 & 71.3 & 91.4\newline
> & \text{Oracle Verifier} & 94.4 & 99.1 & 97.0 & 96.0 & 97.3 & 89.8 & 95.6 \newline
> \hline
>  & \text{Self-Consistency} &  91.4 & 98.5 & 94.7 & 93.4  & 96.2 & 75.2 & 91.6 \newline
> \text{GPT-4+ComplexCoT} & \text{FOBAR} & 91.9 & 98.6 & 94.7 & 94.4 & 96.4 & 75.2 & 91.9\newline
> & \text{Oracle Verifier} & 94.4 & 98.8 & 96.3 & 95.7 & 98.5 & 91.3 & 95.8 \newline
> \hline
> \end{array}
>
> ---
>
> ## **References**
>
> [1] Jason Wei, Xuezhi Wang, Dale Schuurmans, Maarten Bosma, Ed Chi, Brian Ichter, Fei Xia, Quoc Le, and Denny Zhou. Chain of thought prompting elicits reasoning in large language models. Neural Information Processing Systems, 2022.
>
> [2] Srivastava et al. (400+ authors). Beyond the imitation game: Quantifying and extrapolating the capabilities of language models. Transactions on Machine Learning Research, 2023.
>
> [3] Tianci Xue, Ziqi Wang, Zhenhailong Wang, Chi Han, Pengfei Yu, and Heng Ji. RCoT: Detecting and rectifying factual inconsistency in reasoning by reversing chain-of-thought. Preprint arXiv:2305.11499, 2023.
>
> [4] Yixuan Weng, Minjun Zhu, Shizhu He, Kang Liu, and Jun Zhao. Large language models are reasoners with Self-Veriﬁcation. Preprint arXiv:2212.09561, 2022.
>
> [5] Zhan Ling, Yunhao Fang, Xuanlin Li, Zhiao Huang, Mingu Lee, Roland Memisevic, and Hao Su. Deductive veriﬁcation of Chain-of-Thought reasoning. In Neural Information Processing Systems, 2023.

---

> > ### Comment · Reviewer_nUu4 · 2023-11-21
> > **Author response**
> >
> > Thank you for the additional analyses and responses. I think the results with the Oracle experiment are quite promising, and they help justify this line of work. It's also nice to see that there are still some gains on top of Natural Programs, albeit small. It could be interesting to look at inference cost as well, since FOBAR might be less expensive.
> >
> > I thus would like to maintain my score - I think the paper adds a simple and nice idea to this line of work on mathematical reasoning with language models and shows enough results.

---

> > > ### Author Response · Authors · 2023-11-21
> > > **Thank you!**
> > >
> > > We are glad that our reply resolved your concerns.
> > > Thanks again for your positive comments.

---

### Official Review · Reviewer_YtZF · 2023-10-30

**Soundness:** 2 fair
**Presentation:** 3 good
**Contribution:** 2 fair
**Rating:** 3
**Confidence:** 5

**Summary:**

The authors suggest integrating backward reasoning into answer verification to improve the performance of LLMs in mathematical reasoning tasks. In this approach, a number in the original question is masked and replaced with a variable, prompting the LLM to determine the value of x based on a candidate answer. If a candidate answer is correct, the LLM should accurately predict the masked number given this answer. By combining forward and backward reasoning, the authors demonstrate performance improvements across multiple reasoning tasks.

**Strengths:**

1. The paper is well-written and easy to follow.
2. The authors propose using backward reasoning to verify the correctness of the candidate answers.
3. The authors propose using FOBAR, which combines forward and backward reasoning to select the best answer from all the candidates, and they show improvements in experimental results.

**Weaknesses:**

1. Does the proposed backward reasoning have the potential for extension to more complex settings? For some questions, given a candidate number, there can be multiple correct values for the masked number in the question statement. The model may output a number that makes sense, even if it's different from the number in the original question statement. Then, how can we measure the accuracy of the backward reasoning?
2. Can backward reasoning be more accurate than forward reasoning? If backward reasoning isn't simpler than forward reasoning, poor performance in backward reasoning could negatively impact the accuracy of the final answer.

**Questions:**

Recent work proposes using step-by-step verification for answer verification. The paper also suggests backward reasoning, which involves step-by-step analysis. Is there any relationship between backward reasoning and step-by-step verification?

Related literature for step-by-step verification:

Let's Verify Step by Step

Deductive Verification of Chain-of-Thought Reasoning

---

> ### Author Response · Authors · 2023-11-18
> **Reply to Reviewer YtZF ((1/2))**
>
> Thank you for your thoughtful review and valuable feedback. We address your concerns as follows.
>
> > Q1. Does the proposed backward reasoning have the potential for extension to more complex settings? For some questions, given a candidate number, there can be multiple correct values for the masked number in the question statement. The model may output a number that makes sense, even if it's different from the number in the original question statement. Then, how can we measure the accuracy of the backward reasoning?
>
> **A1.** Yes, our proposed backward reasoning **still works** for the mentioned complex setting.
> In backward reasoning, we select the candidate answer that leads to the most backward reasoning chains that predict the masked number successfully. **When there are multiple correct values for the masked number, the proposed backward reasoning can work as long as the correct candidate answer is easier for the LLM to predict the masked number than the wrong candidate answers.** We give an example below.
>
> *Question*: Emily and Sophia are sisters. Emily is 30 years old, and Sophia is 43 years old. Calculate the square of their age difference.
>
> *Candidate Answers*: 169 (correct), 139 (wrong)
>
> *Backward question for the candidate answer 169*: Emily and Sophia are sisters. Emily is 30 years old, and Sophia is ${\bf x}$ years old. Calculate the square of their age difference. If we know the answer to the above question is **169**, what is the value of unknown variable ${\bf x}$?
>
> *Backward question for the candidate answer 139*: Emily and Sophia are sisters. Emily is 30 years old, and Sophia is ${\bf x}$ years old. Calculate the square of their age difference. If we know the answer to the above question is **139**, what is the value of unknown variable ${\bf x}$?
>
> Obviously, for the correct candidate answer $169$, the correct value of ${\bf x}$ is not unique, can be 43 or 17. The proposed backward reasoning for verification still works as the LLM is more likely to predict ${\bf x} = 43$ given the candidate answer $169$ than given $139$. To validate it, we conduct an experiment as follows. For each candidate answer, we sample 100 reasoning chains to predict ${\bf x}$. For the candidate answer 169, there are 24 backward reasoning chains that predict ${\bf x}=43$ (35 chains lead to ${\bf x}=17$, and the remaining 41 chains lead to either 43 or 17 is ok). However, for the candidate answer 139, none of the backward reasoning chains predict ${\bf x}=43$, thus, the candidate answer 169 wins. Hence, in the cases where the correct value of ${\bf x}$ is not unique, the proposed backward reasoning method still works for verification.
>
> ---
>
> > Q2. "Can backward reasoning be more accurate than forward reasoning? If backward reasoning isn't simpler than forward reasoning, poor performance in backward reasoning could negatively impact the accuracy of the final answer."
>
> **A2.** (i) In Section 4.3 of the submission, we conducted an ablation experiment to compare the performance of forward and backward reasoning. As shown in Table 2 there, backward reasoning is more accurate than forward reasoning in 5 of 6 settings.
>
> (ii) It is possible that backward reasoning tasks can be complex. However, **even when the masked number is difficult to predict, as long as the correct candidate answer can better predict the masked number than the incorrect candidate answers, this correct candidate will still be picked.** In other words, the correct candidate answer may still lead to a low prediction accuracy on the masked number, but as long as its accuracy is still higher than those of the incorrect candidate answers, the proposed backward reasoning procedure still works.
>
> (iii) By combining forward and backward reasoning, the proposed **FOBAR is a general method to improve forward verification**. As shown in Table 2 of the submission, FOBAR consistently performs better than Self-Consistency, which uses forward reasoning alone.

---

> ### Author Response · Authors · 2023-11-18
> **Reply to Reviewer YtZF ((2/2))**
>
> > Q3. "Recent work proposes using step-by-step verification for answer verification. The paper also suggests backward reasoning, which involves step-by-step analysis. Is there any relationship between backward reasoning and step-by-step verification? Related literature for step-by-step verification: (i) Let's Verify Step by Step. (ii) Deductive Verification of Chain-of-Thought Reasoning."
>
> A3. (i) Note that the mentioned papers [1, 2] focus on verifying the steps of forward reasoning chains and **DO NOT** use backward reasoning in verification. In [1], the authors label exclusively steps of forward reasoning chains generated by LLMs. The labeled data are then used to train an LLM for verification. Compared with [1], which is computationally expensive in training an LLM and labor-intensive in labeling data, our backward reasoning is training-free for verification and requires no additional data annotation.
>
> (ii) In [2], the authors propose a natural language-based deductive reasoning format that allows the LLM to verify **forward** reasoning steps. In our paper, we use **backward** reasoning to **verify the candidate answers** instead of the steps in forward chains. As backward and forward reasoning are complementary, our backward verification can be combined with their step-by-step forward verifications. We replace the forward verification in our FOBAR (Eq. (4) in our paper) with step-by-step forward verification proposed by [2], and conducted additional experiments on AddSub, GSM8K, and AQuA using GPT-3.5-Turbo. Table below shows the testing accuracy. As can be seen, **combining backward verification with their forward verifications consistently boosts performance**.
>
> \begin{array}{lccc}
> \hline
> &  \text{AddSub} & \text{GSM8K} & \text{AQuA} \newline
> \hline
> \text{NP [2]} & 93.67 &  87.05 & 70.34 \newline
> \text{NP + Backward Verification} & \textbf{93.92} & \textbf{87.89} & \textbf{71.65} \newline
> \hline
> \text{NP + DV + UPV [2]} & 93.54 & 86.01 & 69.49  \newline
> \text{NP + DV + UPV + Backward Verification} & \textbf{93.92} &  \textbf{87.19} & \textbf{70.86} \newline
> \hline
> \end{array}
>
> ---
>
> ## **References**
>
> [1] Hunter Lightman, Vineet Kosaraju, Yura Burda, Harri Edwards, Bowen Baker, Teddy Lee, Jan Leike, John Schulman, Ilya Sutskever, and Karl Cobbe. Let’s verify step by step. Preprint arXiv:2305.20050, 2023.
>
> [2] Zhan Ling, Yunhao Fang, Xuanlin Li, Zhiao Huang, Mingu Lee, Roland Memisevic, and Hao Su. Deductive veriﬁcation of Chain-of-Thought reasoning. In *Neural Information Processing Systems*, 2023.

---

> > ### Comment · Reviewer_YtZF · 2023-11-19
> > **Thanks for the response!**
> >
> > Thank you for your detailed experiments and explanations. However, I still have unresolved concerns regarding the efficacy of backward reasoning, especially about its applicability to more complex problems.
> >
> > 1. The example provided in A1 does not fully convince me. I am curious about the process used to generate the two candidate answers mentioned. Were they generated by language models? Additionally, in broader scenarios, such as in a reasoning sequence A->B, forward reasoning involves generating B given A. In contrast, backward reasoning requires generating A when conditioned on B. However, there could be an infinite number of conditions, like C, where C->A->B or C->B(C$\not=$A). Then, how to judge the correctness of the predicted condition, which can be C instead of A?
> >
> > 2. In A2, you claim that backward reasoning is more accurate for correct candidate answers than for incorrect ones. Could you provide experimental evidence to support this claim? Specifically, what is the accuracy of backward reasoning in both scenarios?

---

> > > ### Author Response · Authors · 2023-11-20
> > > **Reply to Further Comments**
> > >
> > > Thanks for your further comments.
> > >
> > > > Q4. The example provided in A1 does not fully convince me. I am curious about the process used to generate the two candidate answers mentioned. Were they generated by language models?
> > >
> > > **A4.** Yes, candidate answers are generated by language models using Self-Consisntecy, i.e., feed the question into language models and sample reasoning chains from the language models. **The simple example** given in A1 aims to explain that our backward verification can work for suggested cases (i.e., correct values of ${\bf x}$ are not unique).
> > >
> > > ---
> > >
> > > > Q5. Additionally, in broader scenarios, such as in a reasoning sequence A->B, forward reasoning involves generating B given A. In contrast, backward reasoning requires generating A when conditioned on B. However, there could be an infinite number of conditions, like C, where C->A->B or C->B (C$\neq$A). Then, how to judge the correctness of the predicted condition, which can be C instead of A?
> > >
> > >
> > > **A5.** We think there might be some misunderstandings.
> > > The final goal of backward reasoning is to **evaluate the correctness of candidate answers generated by forward reasoning, instead of judging the correctness of the predicted condition**.
> > > For the example given in A1, both 17 and 43 are correct for ${\bf x}$; However, to verify the correctness of candidate answers (i.e., 169 and 139), we only count the number of times that ${\bf x}=43$ is predicted.
> > > For candidate answer 169, we sample 100 backward reasoning chains to predict ${\bf x}$,  24 chains can lead to ${\bf x}=43$ (though 35 chains are leading to 17, which are also correct, we do not count them, as in step 11 of Algorithm 1 in the submission).
> > > For the candidate answer 139, **none of the chains can reach ${\bf x}=43$**. Hence, the candidate answer 169 is better than 139.
> > > Indeed, when the predicted ${\bf x}$ is not equal to the masked number, we do **NOT** need to verify the correctness of ${\bf x}$ (e.g.,  whether ${\bf x}=17$ can lead to the ground-truth answer 169). As in step 11 of Algorithm 1, we only need to check whether the predicted ${\bf x}$ is the masked number using a simple string comparison. In other words, when there are other correct conditions (e.g., ${\bf x}=17$), they are not counted in verifying candidate answers generated by Self-Consistency.
> > >
> > >
> > > ---
> > >
> > > > Q6. In A2, you claim that backward reasoning is more accurate for correct candidate answers than for incorrect ones. Could you provide experimental evidence to support this claim? Specifically, what is the accuracy of backward reasoning in both scenarios?
> > >
> > > **A6.** We think there might be some misunderstandings. Accuracy of backward reasoning is **the accuracy of using backward reasoning in verifying candidate answers generated by Self-Consistency (e.g., 169 and 139 in the Example given in A1)**, **NOT** the accuracy of the predicted ${\bf x}$ being correct (which may not be unique, e.g., ${\bf x}$ can be 43 or 17). **These two accuracies are totally different**. The former is crucial to verify the candidate answers and we only need a simple string comparison (as in step 11 of Algorithm 1). However, computing the latter accuracy is very hard as we must verify the correctness of all predicted ${\bf x}$ (which is challenging).
> > >
> > > The accuracy of using backward reasoning in verifying candidate answers generated by Self-Consistency, as mentioned in A2, is in
> > > Table 2 of the submission. As shown there, in 5 of 6 settings, backward reasoning is more accurate than forward reasoning in verification.

---

> ### Author Response · Authors · 2023-11-21
> **Any other concerns**
>
> Dear Reviewer YtZF,
>
> We would like to thank you again for your detailed reviews.
> We hope that we have satisfactorily addressed your concerns.
>
> Given that your current score is 3, we would be grateful if you could let us know whether our response has addressed your concerns or if you still have any other questions.
>
> We would be happy to do any follow-up discussion or address any additional comments.
>
> Best,
>
> The authors

---

> > ### Comment · Reviewer_YtZF · 2023-11-22
> > **My concerns are still here**
> >
> > Thanks for the explanations and efforts. However, my concerns remain unresolved by your responses.
> >
> > > For Q4: I am asking if the candidate answer in the specific example is generated using LLMs, rather than the general setting. This is because when I tried with ChatGPT, it was able to provide correct answers.
> >
> > Question: Emily and Sophia are sisters. Emily is 30 years old, and Sophia is 43 years old. Calculate the square of their age difference.
> > Candidate Answers: 169 (correct), 139 (wrong)
> > Backward question for the candidate answer 169: Emily and Sophia are sisters. Emily is 30 years old, and Sophia is x years old. Calculate the square of their age difference. If we know the answer to the above question is 169, what is the value of the unknown variable x?
> > Backward question for the candidate answer 139: Emily and Sophia are sisters. Emily is 30 years old, and Sophia is x years old. Calculate the square of their age difference. If we know the answer to the above question is 139, what is the value of the unknown variable x?
> >
> > > For Q5: I am inquiring about general settings, not the specific case you provided. This is because for challenging problems like mathematical proofs, one can always replace the condition with a stronger one to reach the same conclusion. How does your method address this issue? (This is actually a major concern. If the method is only suitable for very simple cases, how can it be extended to more complex settings? Otherwise, the work is not as impressive and may not even achieve a borderline score.)
> >
> > > For Q6: It is necessary to show the **numbers** here. I understand that backward reasoning may improve forward reasoning. However, the performance of backward reasoning also needs to be reported. As mentioned in your response to `A5`, you have a **simple string comparison**, which makes obtaining a metric for backward reasoning possible.

---

> > > ### Author Response · Authors · 2023-11-23
> > > **A real-world example for the complex scenario (non-unique correct values for x)**
> > >
> > > We give an example for the mentioned complex scenarios (multiple correct values for ${\bf x}$) using a question from the testing set of GSM8K (Line 334 at [https://github.com/openai/grade-school-math/blob/master/grade_school_math/data/test.jsonl#L334](https://github.com/openai/grade-school-math/blob/master/grade_school_math/data/test.jsonl#L334)).
> > >
> > > **Question:** `Robert wants to buy a bookshelf and a sofa and has received two offers. Furniture United's offer includes a $1,350 advance payment and 6 monthly installments of $350 each. Cozy Homes' offer includes a $1,100 advance payment and 9 monthly installments of $250 each. What is the cost difference between the two offers?`
> > >
> > > **Ground-truth answer 100:** `Furniture United's offer is 1,350 + (6 *350) = $<<1350+(6*350)=3450>>3,450\nCozy Homes's offer is 1,100 + (9 * 250) = $<<1100+9*250=3350>>3,350\nThe difference is 3,450 – 3,350 = $<<3450-3350=100>>100\n#### 100`
> > >
> > > **Candidate answers generated by Self-Consistency**: 100 (4 times), 200 (6 times); the forward reasoning chains are generated by CoT + GPT-3.5-Turbo with temperature 0.7, we attach them in the next reply.
> > >
> > >
> > > **Backward question for the candidate answer 100**:
> > > `Robert wants to buy a bookshelf and a sofa and has received two offers. Furniture United's offer includes a x advance payment and 6 monthly installments of $350 each. Cozy Homes' offer includes a $1,100 advance payment and 9 monthly installments of $250 each. What is the cost difference between the two offers? If we know the answer to the above question is 100, what is the value of unknown variable x?`
> > >
> > > We sample 8 backward reasoning chains to predict ${\bf x}$: **1150** (1 times) and **1350** (7 times). Note that 1150 and 1350 are correct values for ${\bf x}$ for the above backward question. The chains are attached in the below replies.
> > >
> > > **Backward question for the candidate answer 200**: `Robert wants to buy a bookshelf and a sofa and has received two offers. Furniture United's offer includes a x advance payment and 6 monthly installments of $350 each. Cozy Homes' offer includes a $1,100 advance payment and 9 monthly installments of $250 each. What is the cost difference between the two offers? If we know the answer to the above question is 200, what is the value of unknown variable x?`
> > >
> > > We sample 8 backward reasoning chains to predict ${\bf x}$: **1450** (5 times) and **1050** (3 times). Note that 1450 and 1050 are correct values for ${\bf x}$ for the above backward question. The chains are attached in the below replies.
> > >
> > > **Hence, the candidate answer 100 wins, as it has 7 backward reasoning chains reaching 1350, while the candidate answer 200 has no backward chains reaching 1350.**

---

> ### Author Response · Authors · 2023-11-23
> **Reply to remaining concerns (1/2)**
>
> Thanks for your further comments and clarification of previous concerns
>
> > Q7. For Q4: I am asking if the candidate answer in the specific example is generated using LLMs, rather than the general setting. This is because when I tried with ChatGPT, it was able to provide correct answers.
>
> **A7.** We agree ChatGPT will provide the correct answer to the simple question (*Emily and Sophia are sisters. Emily is 30 years old, and Sophia is 43 years old. Calculate the square of their age difference*) given in A1. However, this example is designed to showcase how our algorithm works when ${\bf x}$ has multiple correct values. We apologize for the confusion caused by this example. As searching for such real-world examples from datasets is labor-intensive. Please give us some time, and we will attach examples here later.
>
> ---
>
> > Q8. For Q5 `(Additionally, in broader scenarios, such as in a reasoning sequence A->B, forward reasoning involves generating B given A. In contrast, backward reasoning requires generating A when conditioned on B. However, there could be an infinite number of conditions, like C, where C->A->B or C->B (C
> A). Then, how to judge the correctness of the predicted condition, which can be C instead of A?)`: I am inquiring about general settings, not the specific case you provided. This is because for challenging problems like mathematical proofs, one can always replace the condition with a stronger one to reach the same conclusion. How does your method address this issue? (This is actually a major concern. If the method is only suitable for very simple cases, how can it be extended to more complex settings? Otherwise, the work is not as impressive and may not even achieve a borderline score.)
>
>
> **A8.** Thanks for clarifying previous concerns. Our FOBAR algorithm is general and capable of handling the scenarios given in Q5. Indeed, FOBAR only needs to check whether the prediction of backward chains is exactly the same as the masked number by string comparison and does not need to "*judge the correctness of the predicted condition*". We show how Algorithm 1 (step 5 - step14) in the submission applies to  the mentioned scenario (existing C s.t. `C->A->B`) as follows (indeed, has no difference between complex scenarios (multiple correct ${\bf x}$'s) and simple scenarios (unique correct ${\bf x}$)):
>
> - let $\\{B\_i\\}\_{i=1}^{T}$ be $T$ candidate answers obtained by Self-Consistency, calculate $\mathbb{P}\_{\text{forward}}(B_i)$;
> - for each candidate answer $B\_i \in \\{B\_1, \dots, B\_T\\}$:
>     - mask the number $A$ in the question, sample $M\_B$ backward chains with predictions $\{\hat{A}\_{B\_i}^{(n)}\}\_{n=1}^{M_B}$;
>     - **count number of correct prediction ${\bf Z\_{B\_{i}} = \sum_{n=1}^{M\_B} \mathbb{I}(\hat{A}\_{B\_i}^{(n)}=A)}$;**
> - compute probability: for $B\_i \in \\{B\_1, \dots, B\_T\\}$, calculate
> $${\bf \mathbb{P}\_{\text{backward}}(B\_i)=\frac{Z\_{B\_i}+\epsilon}{\sum\_{j=1}^TZ\_{B\_j} + T\epsilon}},$$
> where $\epsilon$ is a small positive constant;
> - combining forward and backward probability: for $B\_i \in \\{B\_1, \dots, B\_T\\}$, calculate
> $$\mathbb{P}(B_i)\propto (\mathbb{P}\_{\text{forward}}(B\_i))^{\alpha}(\mathbb{P}\_{\text{backward}}(B\_i))^{1-\alpha},$$
> where $\alpha=0.5$.
>
> For the **more complex scenarios** (an infinite number of values are correct for the masked position), there may not be a single backward reasoning chain that reaches the masked number exactly, and we will get $Z\_{B\_{i}}=0$ for all $B\_i \in \\{B\_1, \dots, B\_T\\}$, therefore, **backward verification fails** since $\mathbb{P}\_{\text{backward}}(B\_i)=\epsilon$ for all $B_{i}$. Our FOBAR, which combines forward and backward reasoning, becomes Self-Consistency, i.e., $$\mathbb{P}(B\_i)\propto (\mathbb{P}\_{\text{forward}}(B\_i))^{\alpha}(\mathbb{P}\_{\text{backward}}(B\_i))^{1-\alpha}=(\mathbb{P}\_{\text{forward}}(B\_i))^{\alpha}.$$
>
> We hope our reply can resolve your concerns about how to apply our backward verification and FOBAR to complex scenarios.

---

> ### Author Response · Authors · 2023-11-23
> **Reply to remaining concerns (2/2)**
>
> > Q9. For Q6: It is necessary to show the numbers here. I understand that backward reasoning may improve forward reasoning. However, the performance of backward reasoning also needs to be reported. As mentioned in your response to A5, you have a simple string comparison, which makes obtaining a metric for backward reasoning possible.
>
> **A9.** Thanks for giving us the opportunity to show the numbers to support an intuition "**the correct candidate answer can predict the masked number **more accurately** than the incorrect candidate answers**". We calculate two metrics (averaged over all backward questions across six tasks) in backward reasoning:
> - $\mathbb{P}({\bf x}={\text{the masked number}}|\text{correct candidate answer})=\frac{\text{\\#\\{backward chains with the correct candidate answer that predict the masked number exactly\\}}}{\text{\\#\\{backward chains with the correct candidate answer\\}}}$
> - $\mathbb{P}({\bf x}={\text{the masked number}}|\text{incorrect candidate answers})=\frac{\text{\\#\\{backward chains with the incorrect candidate answers that predict the masked number exactly\\}}}{\text{\\#\\{backward chains with the incorrect candidate answers\\}}}$
>
> Table below shows the results.
> We can see that the correct candidate answer can predict the masked number **much more accurately** than the incorrect candidate answers in predicting the masked number.
> This is crucial evidence for explaining why backward verification works. We have added Figure 5 and a new Subsection 4.5 in the updated paper. We deeply appreciate the reviewer's suggestion.
>
> \begin{array}{l| c| c|c}
> \hline
> &
> \text{text-davinci-003} &
> \text{GPT-3.5-Turbo} & \text{GPT-4} \newline
> & \text{CoT}\quad         \text{ComplexCoT} &   \text{CoT}\quad         \text{ComplexCoT} & \text{CoT}\quad         \text{ComplexCoT}\newline
> \hline
> \mathbb{P}({\bf x}={\text{the masked number} | \text{incorrect candidate answer}})   & 13.55\quad   \quad\quad\quad  8.59 & 12.79 \quad   \quad\quad\quad  11.25 &  25.03 \quad\quad\quad\quad 19.24\newline
> \mathbb{P}({\bf x}={\text{the masked number} | \text{correct candidate answers} })
>  & 41.26\quad   \quad\quad\quad  48.01 & 41.58 \quad   \quad\quad\quad 42.08  & 47.18  \quad\quad\quad\quad 53.00\newline
> \hline
> \end{array}
>
>
> If there is any additional explanation or experiments that can save the reviewer’s time to understand our paper and clarify the concerns, we will be more than happy to do so.

---

> ### Author Response · Authors · 2023-11-23
> **Forward reasoning chains**
>
> The below 10 forward reasoning chains are generated by Cot + GPT-3.5-Turbo, temperate=0.7, demonstrations are the same as Table 10 in the CoT paper (https://arxiv.org/abs/2201.11903), as attached in the next replies.
>
> ```
> chain 1: Furniture United's offer includes a $1,350 advance payment and 6 monthly installments of $350 each. So the total cost of their offer is 1,350 + 6 * 350 = $3,450.
> Cozy Homes' offer includes a $1,100 advance payment and 9 monthly installments of $250 each. So the total cost of their offer is 1,100 + 9 * 250 = $3,350.
> The cost difference between the two offers is 3,450 - 3,350 = $100. The answer is $100.
>
> chain 2: Furniture United's offer includes an advance payment of $1,350 and 6 monthly installments of $350 each. So the total cost for Furniture United's offer is 1350 + (6 * 350) = 1350 + 2100 = $3,450.
> Cozy Homes' offer includes an advance payment of $1,100 and 9 monthly installments of $250 each. So the total cost for Cozy Homes' offer is 1100 + (9 * 250) = 1100 + 2250 = $3,350.
> The cost difference between the two offers is 3450 - 3350 = $100. The answer is $100.
>
> chain 3: Furniture United's offer includes a $1,350 advance payment and 6 monthly installments of $350 each. So the total cost of their offer is 1,350 + (6 * 350) = 1,350 + 2,100 = $3,450.
> Cozy Homes' offer includes a $1,100 advance payment and 9 monthly installments of $250 each. So the total cost of their offer is 1,100 + (9 * 250) = 1,100 + 2,250 = $3,350.
> The cost difference between the two offers is 3,450 - 3,350 = $100. The answer is $100.
>
> chain 4: Furniture United's offer costs $1350 + 6 * $350 = $3450 dollars in total. Cozy Homes' offer costs $1100 + 9 * $250 = $3350 dollars in total.
> So the cost difference between the two offers is $3450 - $3350 = 100 dollars. The answer is 100.
>
> chain 5: For Furniture United's offer, the total cost is $1,350 (advance payment) + 6 * $350 (monthly installments) = $3,150.
> For Cozy Homes' offer, the total cost is $1,100 (advance payment) + 9 * $250 (monthly installments) = $3,350.
> The cost difference between the two offers is $3,350 - $3,150 = $200. The answer is $200.
>
> chain 6: Furniture United's offer includes a $1,350 advance payment and 6 monthly installments of $350 each, so the total cost is 1,350 + (6 * 350) = $3,150.
> Cozy Homes' offer includes a $1,100 advance payment and 9 monthly installments of $250 each, so the total cost is 1,100 + (9 * 250) = $3,350.
> The cost difference between the two offers is 3,350 - 3,150 = $200. The answer is $200.
>
> chain 7: Furniture United's offer includes a $1,350 advance payment and 6 monthly installments of $350 each. So the total cost of Furniture United's offer is 1350 + (6 * 350) = $3,150.
> Cozy Homes' offer includes a $1,100 advance payment and 9 monthly installments of $250 each. So the total cost of Cozy Homes' offer is 1100 + (9 * 250) = $3,350.
> The cost difference between the two offers is 3350 - 3150 = $200. The answer is $200.
>
> chain 8: Furniture United's offer includes a $1,350 advance payment and 6 monthly installments of $350 each. So the total cost for Furniture United's offer is 1350 + (6 * 350) = $3,150.
> Cozy Homes' offer includes a $1,100 advance payment and 9 monthly installments of $250 each. So the total cost for Cozy Homes' offer is 1100 + (9 * 250) = $3,350.
> The cost difference between the two offers is 3350 - 3150 = $200. The answer is $200.
>
> chain 9: Furniture United's offer includes a $1,350 advance payment and 6 monthly installments of $350 each. So the total cost for Furniture United's offer is 1350 + (6 * 350) = $3,150.
> Cozy Homes' offer includes a $1,100 advance payment and 9 monthly installments of $250 each. So the total cost for Cozy Homes' offer is 1100 + (9 * 250) = $3,350.
> The cost difference between the two offers is 3350 - 3150 = $200. The answer is $200.
>
> chain 10: Furniture United's offer includes a $1,350 advance payment and 6 monthly installments of $350 each. So the total cost for Furniture United's offer is 1350 + (6 * 350) = $3,150.
> Cozy Homes' offer includes a $1,100 advance payment and 9 monthly installments of $250 each. So the total cost for Cozy Homes' offer is 1100 + (9 * 250) = $3,350.
> The cost difference between the two offers is 3350 - 3150 = $200. The answer is $200.
> ```

---

> ### Author Response · Authors · 2023-11-23
> **Backward reasoning chains for the candidate answer 100**
>
> ```
> ### chain 1
> ```Let's think step by step.
> Furniture United's offer includes a $x advance payment and 6 monthly installments of $350 each, so the total cost of their offer is x + 6 * $350 = x + $2100.
> Cozy Homes' offer includes a $1,100 advance payment and 9 monthly installments of $250 each, so the total cost of their offer is $1,100 + 9 * $250 = $1,100 + $2250 = $3350.
> The cost difference between the two offers is: (x + $2100) - $3350 = x - $1250.
> We are given that the cost difference is $100, so we can write: x - $1250 = $100.
> Solving for x, we get: x = $1350.
> The value of x is $1350.
>
> ### chain 2
> Let's think step by step.
> Furniture United's offer includes an advance payment of $x and 6 monthly installments of $350 each. So the total cost of Furniture United's offer is x + 6 * $350 = x + $2100.
> Cozy Homes' offer includes an advance payment of $1,100 and 9 monthly installments of $250 each. So the total cost of Cozy Homes' offer is $1100 + 9 * $250 = $1100 + $2250 = $3350.
> The cost difference between the two offers is the difference between the total costs: ($x + $2100) - $3350.
> We are given that the cost difference is $100, so we can write: ($x + $2100) - $3350 = $100.
> Simplifying the left side, we get: $x + $2100 - $3350 = $100.
> Combining like terms, we get: $x - $1250 = $100.
> Adding $1250 to both sides, we get: $x = $1350.
> The value of x is $1350.
>
> ### chain 3
> Let's think step by step.
> Furniture United's offer includes a $x advance payment and 6 monthly installments of $350 each. So the total cost of Furniture United's offer is x + 6 * $350 = x + $2100.
> Cozy Homes' offer includes a $1,100 advance payment and 9 monthly installments of $250 each. So the total cost of Cozy Homes' offer is $1100 + 9 * $250 = $1100 + $2250 = $3350.
> The cost difference between the two offers is the difference between the total costs: ($x + $2100) - $3350 = $x - $1250.
> We are given that the cost difference is $100, so we can write: $x - $1250 = $100.
> Adding $1250 to both sides, we get: $x = $1350.
> The value of x is $1350.
>
> ### chain 4
> Let's think step by step.
> Furniture United's offer includes a $x advance payment and 6 monthly installments of $350 each. So the total cost of Furniture United's offer is x + 6 * $350.
> Cozy Homes' offer includes a $1,100 advance payment and 9 monthly installments of $250 each. So the total cost of Cozy Homes' offer is $1,100 + 9 * $250.
> The cost difference between the two offers is the difference between the total costs: (x + 6 * $350) - ($1,100 + 9 * $250).
> We are given that the cost difference is $100, so we can write: (x + 6 * $350) - ($1,100 + 9 * $250) = $100.
> Simplifying the left side, we get: x + $2,100 - ($1,100 + $2,250) = $100.
> Simplifying further, we get: x + $2,100 - $3,350 = $100.
> Combining like terms, we get: x - $1,250 = $100.
> Adding $1,250 to both sides, we get: x = $1,350.
> The value of x is $1,350.
>
> ### chain 5
> Let's think step by step.
> Furniture United's offer includes a $x advance payment and 6 monthly installments of $350 each. So the total cost of Furniture United's offer is: x + 6 * $350 = x + $2100.
> Cozy Homes' offer includes a $1,100 advance payment and 9 monthly installments of $250 each. So the total cost of Cozy Homes' offer is: $1,100 + 9 * $250 = $1,100 + $2250 = $3,350.
> The cost difference between the two offers is: (x + $2100) - $3,350 = x - $1,250.
> We are given that the cost difference is $100, so we can write: x - $1,250 = $100.
> Solving for x, we get: x = $1,350.
> The value of x is $1,350.
>
> ### chain 6
> Let's think step by step.
> Furniture United's offer includes a $x advance payment and 6 monthly installments of $350 each. So the total cost of Furniture United's offer is x + 6 * $350 = x + $2100.
> Cozy Homes' offer includes a $1,100 advance payment and 9 monthly installments of $250 each. So the total cost of Cozy Homes' offer is $1100 + 9 * $250 = $1100 + $2250 = $3350.
> The cost difference between the two offers is: (x + $2100) - $3350 = x - $1250.
> We are given that the cost difference is $100, so we can write: x - $1250 = $100.
> Solving for x, we get: x = $1350.
> The value of x is $1350.
> ```

---

> ### Author Response · Authors · 2023-11-23
> **Backward reasoning chains for candidate answer 100 (Cont')**
>
> ```
> ### chain 7
> Let's think step by step.
> Furniture United's offer includes a $x advance payment and 6 monthly installments of $350 each. The total cost of Furniture United's offer can be calculated as: x + 6 * $350 = x + $2,100.
> Cozy Homes' offer includes a $1,100 advance payment and 9 monthly installments of $250 each. The total cost of Cozy Homes' offer can be calculated as: $1,100 + 9 * $250 = $1,100 + $2,250 = $3,350.
> The cost difference between the two offers is: (x + $2,100) - $3,350 = x + $2,100 - $3,350 = x - $1,250.
> We are given that the cost difference is $100, so we can write: x - $1,250 = $100.
> Solving for x, we get: x = $1,350.
> The value of x is $1,350.
>
> ### chain 8
> Let's think step by step.
> Furniture United's offer includes a $x advance payment and 6 monthly installments of $350 each. So the total cost for Furniture United's offer is x + (6 * $350) = x + $2100.
> Cozy Homes' offer includes a $1,100 advance payment and 9 monthly installments of $250 each. So the total cost for Cozy Homes' offer is $1100 + (9 * $250) = $1100 + $2250 = $3350.
> The cost difference between the two offers is the difference between the total costs: ($3350) - (x + $2100) = $3350 - $2100 - x = $1250 - x.
> We are given that the cost difference is $100, so we can write: $1250 - x = $100.
> Solving for x, we get: x = $1150.
> The value of x is $1150.
> ```

---

> ### Author Response · Authors · 2023-11-23
> **Backward reasoning chains for candidate answer 200**
>
> ```
> ### chain 1
> Let's think step by step.
> Furniture United's offer includes a $x advance payment and 6 monthly installments of $350 each. So the total cost of Furniture United's offer is x + 6 * 350 = x + 2100.
> Cozy Homes' offer includes a $1,100 advance payment and 9 monthly installments of $250 each. So the total cost of Cozy Homes' offer is 1100 + 9 * 250 = 1100 + 2250 = 3350.
> The cost difference between the two offers is the difference between the total costs: (x + 2100) - 3350 = x + 2100 - 3350 = x - 1250.
> We are given that the cost difference is 200, so we can write: x - 1250 = 200.
> Solving for x, we get: x = 1450.
> The value of x is 1450.
>
> ### chain 2
> Let's think step by step.
> Furniture United's offer includes an advance payment of $x and 6 monthly installments of $350 each. So the total cost of their offer is x + 6 * $350 = x + $2100.
> Cozy Homes' offer includes an advance payment of $1,100 and 9 monthly installments of $250 each. So the total cost of their offer is $1100 + 9 * $250 = $1100 + $2250 = $3350.
> The cost difference between the two offers is the total cost of Cozy Homes' offer minus the total cost of Furniture United's offer: $3350 - (x + $2100) = $1250 - x.
> We are given that the cost difference is $200, so we can write: $1250 - x = $200.
> Solving for x, we get: x = $1050.
> The value of x is $1050.
>
> ### chain 3
> Let's think step by step.
> Furniture United's offer includes a $x advance payment and 6 monthly installments of $350 each. So the total cost of their offer is x + 6 * $350 = x + $2100.
> Cozy Homes' offer includes a $1,100 advance payment and 9 monthly installments of $250 each. So the total cost of their offer is $1,100 + 9 * $250 = $1,100 + $2,250 = $3,350.
> The cost difference between the two offers is the difference between the total costs: ($3,350) - (x + $2,100) = $3,350 - $2,100 - x = $1,250 - x.
> We are given that the cost difference is $200, so we can write: $1,250 - x = $200.
> Solving for x, we get: x = $1,250 - $200 = $1,050.
> The value of x is $1,050.
>
> ### chain 4
> Let's think step by step.
> Furniture United's offer includes a $x advance payment and 6 monthly installments of $350 each. So the total cost for Furniture United's offer is x + 6 * $350 = x + $2100.
> Cozy Homes' offer includes a $1,100 advance payment and 9 monthly installments of $250 each. So the total cost for Cozy Homes' offer is $1100 + 9 * $250 = $1100 + $2250 = $3350.
> The cost difference between the two offers is the difference in their total costs: ($x + $2100) - $3350 = $x - $1250.
> We are given that the cost difference is $200, so we can write: $x - $1250 = $200.
> Adding $1250 to both sides, we get: $x = $1450.
> The value of x is $1450.
>
> ### chain 5
> Let's think step by step.
> Furniture United's offer includes a $x advance payment and 6 monthly installments of $350 each. So the total cost of Furniture United's offer is x + 6 * 350 = x + 2100.
> Cozy Homes' offer includes a $1,100 advance payment and 9 monthly installments of $250 each. So the total cost of Cozy Homes' offer is 1100 + 9 * 250 = 1100 + 2250 = 3350.
> The cost difference between the two offers is the difference between the total costs: (x + 2100) - 3350 = x + 2100 - 3350 = x - 1250.
> We are given that the cost difference is 200, so we can write: x - 1250 = 200.
> Solving for x, we get: x = 1450.
> The value of x is 1450.
>
> ### chain 6
> Let's think step by step.
> Furniture United's offer includes a $x advance payment and 6 monthly installments of $350 each. So the total cost for Furniture United's offer is x + 6*350 = x + 2100.
> Cozy Homes' offer includes a $1,100 advance payment and 9 monthly installments of $250 each. So the total cost for Cozy Homes' offer is 1100 + 9*250 = 1100 + 2250 = 3350.
> The cost difference between the two offers is the difference between the total costs: (x + 2100) - 3350 = x - 1250.
> We are given that the cost difference is $200, so we can write: x - 1250 = 200.
> Solving for x, we get: x = 1450.
> The value of x is 1450.
> ```

---

> ### Author Response · Authors · 2023-11-23
> **Backward reasoning chains for candidate answer 200 (Cont')**
>
> ```
> ### chain 7
> Let's think step by step.
> Furniture United's offer includes a $x advance payment and 6 monthly installments of $350 each. So the total cost of their offer is x + 6 * $350 = x + $2100.
> Cozy Homes' offer includes a $1,100 advance payment and 9 monthly installments of $250 each. So the total cost of their offer is $1100 + 9 * $250 = $1100 + $2250 = $3350.
> The cost difference between the two offers is the difference between the total costs: ($x + $2100) - $3350 = $x + $2100 - $3350 = $x - $1250.
> We are given that the cost difference is $200, so we can write: $x - $1250 = $200.
> Adding $1250 to both sides, we get: $x = $200 + $1250 = $1450.
> The value of x is $1450.
>
> ### chain 8
> Let's think step by step.
> Furniture United's offer includes a $x advance payment and 6 monthly installments of $350 each. So the total cost of Furniture United's offer is x + 6 * $350 = x + $2100.
> Cozy Homes' offer includes a $1,100 advance payment and 9 monthly installments of $250 each. So the total cost of Cozy Homes' offer is $1100 + 9 * $250 = $1100 + $2250 = $3350.
> The cost difference between the two offers is: ($1100 + $2250) - (x + $2100) = $3350 - (x + $2100) = $3350 - $2100 - x = $1250 - x.
> We are given that the cost difference is $200, so we can write: $1250 - x = $200.
> Solving for x, we get: x = $1050.
> The value of x is $1050.
> ```

---

> ### Author Response · Authors · 2023-11-23
> **Eight CoT demonstrations**
>
> ```
> Q: There are 15 trees in the grove. Grove workers will plant trees in the grove today. After they are done, there will be 21 trees. How many trees did the grove workers plant today?
> A: There are 15 trees originally. Then there were 21 trees after the Grove workers planted some more. So there must have been 21 - 15 = 6 trees that were planted. The answer is 6.
>
> Q: If there are 3 cars in the parking lot and 2 more cars arrive, how many cars are in the parking lot?
> A: There are originally 3 cars. Then 2 more cars arrive. Now 3 + 2 = 5 cars are in the parking lot. The answer is 5.
>
> Q: Leah had 32 chocolates and her sister had 42. If they ate 35, how many pieces do they have left in total?
> A: Originally, Leah had 32 chocolates and her sister had 42. So in total they had 32 + 42 = 74. After eating 35, they had 74 - 35 = 39 pieces left in total. The answer is 39.
>
> Q: Jason had 20 lollipops. He gave Denny some lollipops. Now Jason has 12 lollipops. How many lollipops did Jason give to Denny?
> A: Jason had 20 lollipops originally. Then he had 12 after giving some to Denny. So he gave Denny 20 - 12 = 8 lollipops. The answer is 8.
>
> Q: Shawn has five toys. For Christmas, he got two toys each from his mom and dad. How many toys does he have now?
> A: Shawn started with 5 toys. He then got 2 toys each from his mom and dad. So he got 2 * 2 = 4 more toys. Now he has 5 + 4 = 9 toys. The answer is 9.
>
> Q: There were nine computers in the server room. Five more computers were installed each day, from monday to thursday. How many computers are now in the server room?
> A: There were originally 9 computers. For each day from monday to thursday, 5 more computers were installed. So 4 * 5 = 20 computers were added. Now 9 + 20 = 29 computers are now in the server room. The answer is 29.
>
> Q: Michael had 58 golf balls. On tuesday, he lost 23 golf balls. On wednesday, he lost 2 more. How many golf balls did he have at the end of wednesday?
> A: Michael started with 58 golf balls. He lost 23 on Tuesday, and lost 2 more on wednesday. So he had 58 - 23 = 35 at the end of Tuesday, and 35 - 2 = 33 at the end of wednesday. The answer is 33.
>
> Q: Olivia has $23. She bought five bagels for $3 each. How much money does she have left?
> A: Olivia had 23 dollars. She bought 5 bagels for 3 dollars each. So she spent 5 * 3 = 15 dollars. Now she has 23 - 15 = 8 dollars left. The answer is 8.
> ```

---

### Official Review · Reviewer_PfVM · 2023-11-01

**Soundness:** 3 good
**Presentation:** 3 good
**Contribution:** 1 poor
**Rating:** 3
**Confidence:** 4

**Summary:**

The authors propose forward-backward reasoning prompting. Once candidate answers are generated for a query, a number is masked in the query (replacing it with “x”), and the model is asked to predict the masked number given the candidate answer. Specifically, the following template is used: “If we know the answer to the above question is ${A_c}$, what is the value of unknown variable x?” The prompt is appended to the query with the masked number and candidate generation. The model is more likely to predict the correct value for “x” if the candidate answer ${A_c}$ is correct.

This method is tested with three LLMs on six arithmetic reasoning tasks. The method is motivated by the observation that improvements from self-consistency, which chooses an answer by majority voting over multiple reasoning chains, plateau as more reasoning chains are sampled. The authors show that their method outperforms forward reasoning alone.

**Strengths:**

The problem is well motivated. The authors show that (average) testing accuracy of self-consistency plateaus as more candidate answers are temperature sampled.

The proposed verification method is straightforward and clear. The proposed verification template does not need to be generated compared to some of the related works.

The authors did a great job with the experiments comparing FOBAR to multiple reasonable baselines.

**Weaknesses:**

Although the proposed template is straightforward, it would have to be modified according to the tasks on which it’s being applied to. This style of verification always requires something to be predicted. Part of the attractiveness of self-consistency is the fact that it can be applied out-of-the-box to any task; however, the proposed method needs to be modified across tasks and type of responses.

The method lacks novelty as it bears a strong resemblance to RCoT which tries to re-generate the question conditioned on the candidate response. FOBAR seems essentially as an extension of the factual consistency check within RCoT. Moreover, the marginal difference in results between both methods further supports this perspective.

Given that the experiments are carried out using OpenAI APIs, which undergo regular updates, it would be advisable to ensure that the baselines are run with the same API snapshots. The slight variance in the scores might be attributed to the differences in the API versions.

**Questions:**

How does the backward reasoning compare to verification of answers? E.g. Once you generate multiple candidate answers using forward reasoning, you can ask a model to “verify” the candidate answer given the whole context (Similar to verifier for GSM8K in Cobbe et al 2021)

Regarding my concern about OpenAI APIs getting updated regularly, have the authors used multiple different seeds for the experiments?

Given some known LLM behaviors and order of chains of arithmetic reasoning, do you think there will be a noticeable difference in performance if only the numbers at the end of the reasoning chains were masked to be predicted compared to numbers at the beginning of the reasoning?

Do you have any insights on how the verification method would perform on non arithmetic reasoning tasks?

---

> ### Author Response · Authors · 2023-11-18
> **Reply to Reviewer PfVM (1/4)**
>
> Thanks for your thoughtful review and valuable feedback. We address your concerns as follows.
>
> > Q1. "Although the proposed template is straightforward, it would have to be modified according to the tasks on which it’s being applied to. This style of verification always requires something to be predicted. Part of the attractiveness of self-consistency is the fact that it can be applied out-of-the-box to any task; however, the proposed method needs to be modified across tasks and type of responses."
> >
> > "Do you have any insights on how the verification method would perform on non arithmetic reasoning tasks?"
>
> **A1.** (i) In this paper, **our focus is on mathematical tasks**, which often require a number of reasoning steps to reach the answer and are thus challenging. The proposed template (i.e., masking a number in the question and then asking the LLM to predict the masked number) is **simple**, **general**, and **can be applied out-of-the-box to any mathematical task**.
>
> (ii) **Extension to non-arithmetic tasks is possible** and follows a similar line. Similar to that for mathematical tasks, we **mask informative word/token/character** and then ask the LLM to predict. We give **two examples (Date Understanding reasoning task [1, 2] and Last-Letter-Concatenation reasoning task [1])** as follows.
>
> `Example 1: Date Understanding`
>
> *Question*: Yesterday was April 30, 2021. What is the date today in MM/DD/YYYY?
>
> *Candidate Answer*: 05/01/2021 (correct), 05/02/2021 (wrong)
>
> *Backward question for the candidate answer 05/01/2021*:
> Yesterday was April 30, 2021. What is the date **x** in MM/DD/YYYY? If we know the answer to the above question is **05/01/2021**, what is the English word at **x**?
>
> *Backward question for the candidate answer 05/02/2021*:
> Yesterday was April 30, 2021. What is the date **x** in MM/DD/YYYY? If we know the answer to the above question is **05/02/2021**, what is the English word at **x**?
>
> Intuitively, the LLM is more likely to predict the word "today" given the correct candidate answer 05/01/2021. We leave the details as future work.
>
> `Example 2: Last-Letter-Concatenation`
>
> *Question*: Take the last letters of each word in "Whitney Erika Tj Benito" and concatenate them.
>
> *Candidate Answer*: yajo (correct), yaji (wrong)
>
> *Backward question for candidate answer yajo*:
> Take the last letters of each word in "Whitney Erika Tj Benit[]" and concatenate them. If we know the answer to the above question is **yajo**, what is the character at []?
>
> *Backward question for candidate answer yaji*:
> Take the last letters of each word in "Whitney Erika Tj Benit[]" and concatenate them. If we know the answer to the above question is **yaji**, what is the character at []?
>
> Note that we use "[]" to mask the character instead of "x" (which is also a character). The LLM is more likely to predict "o" correctly at [] given the candidate answer yajo than yaji. Hence, the proposed FOBAR can be used in other types of tasks by masking the informative word/token/character in the questions.
>
> (iii) Note that one major contribution of the proposed FOBAR is a **novel** method to combine forward and backward reasoning for verification (i.e., $\mathbb{P}(\hat{A}_c) \propto (\mathbb{P}\_\text{forward}(\hat{A}_c) )^\alpha (\mathbb{P}\_\text{backward}(\hat{A}_c) )^{1-\alpha}$). The proposed method is **general** and **can be integrated into existing verification methods** (such as RCoT [3] and Self-Verification [4]) for non-arithmetic reasoning tasks.

---

> ### Author Response · Authors · 2023-11-18
> **Reply to Reviewer PfVM (2/4)**
>
> > Q2. "The method lacks novelty as it bears a strong resemblance to RCoT which tries to re-generate the question conditioned on the candidate response. FOBAR seems essentially as an extension of the factual consistency check within RCoT. Moreover, the marginal difference in results between both methods further supports this perspective."
>
> **A2.** We believe there might be some misunderstanding. FOBAR is a **novel** method for verification and does **NOT** bear a strong resemblance to RCoT.
>
> (i) RCoT asks the LLM to reconstruct the conditions (a sentence) of the question and candidate answers, while FOBAR asks the LLM to predict a masked number in the question. Predicting a number is **easier** than reconstructing the condition (a sequence of tokens). Furthermore, **FOBAR is a novel method to combine forward and backward reasoning for verification, while RCoT uses the backward procedure alone for verification**.
>
> (ii) To check factual consistency, **RCoT needs three complex steps** ((1) problem decomposition: decompose the original problem $Q$ and reconstructed problem $\hat{Q}$ into a list of and conditions $L_Q$ and $L_{\hat{Q}}$; (2) condition comparison: query the LLM $nm$ times and conduct $mn$ comparisons to check whether their condition lists $L_Q$ and $L_{\hat{Q}}$ are the same, where $m$ and $n$ are the length of $L_Q$ and $L_{\hat{Q}}$, respectively; (3) question comparison: ask the LLM to checking whether the question is misinterpreted). However, FOBAR simply checks whether the masked number is predicted correctly, and is **much simpler and easier**. Moreover, the three steps taken by RCoT are not only complex, but also **costly** since the LLM must be queried $nm$ times. On the other hand, **FOBAR only requires performing inexpensive string comparisons**.
>
> (iii) The complex steps of RCoT in checking factual consistency can cause **inaccurate backward verification**. As can be seen from Table 1 of the submission, **FOBAR consistently performs better than RCoT**. Furthermore, the improvements in testing accuracy are large, e.g., on SVAMP (**+3.6**), GSM8K (**+2.8**), AQuA (**+10.1**).
>
> ---
>
> > Q3. "Given that the experiments are carried out using OpenAI APIs, which undergo regular updates, it would be advisable to ensure that the baselines are run with the same API snapshots. The slight variance in the scores might be attributed to the differences in the API versions."
>
> **A3.** (i) Using the OpenAI APIs for CoT-based research is **standard**, e.g.,  Self-Consistency [5], ComplexCoT [6], Least-to-Most Prompting [7], Tree-of-Thoughts [8].
>
> (ii) We conducted the experiments using OpenAI's APIs between 1 August 2023 and 20 August 2023. The baselines (CoT, ComplexCoT, Self-Consistency, Self-Veriﬁcation) are based on the (roughly) same model snapshots as our FOBAR (**Note that only OpenAI itself can use the exactly same API snapshot**). For the other baselines (PHP, RE2, RCoT), we copy the results reported in their publications, which is **standard** in CoT-based research as querying OpenAI's API is expensive (e.g., RCoT [3], Self-Verification [4], ComplexCoT [6], Zero-Shot CoT [10], PAL [11]).
>
> (iii) **The improvements of FOBAR over baselines are consistent and large**. Table 1 in the submission shows that FOBAR achieves **higher** accuracies over baselines (e.g.,  FOBAR has much higher accuracy than previous SOTA methods on MultiArith (**+4.1**), SVAMP (**+3.7**), and GSM8K (**+2.9**) when using text-davinci-003 with CoT Prompting). These improvements are attributed to FOBAR, which combines forward and backward reasoning for verifying candidate answers.

---

> ### Author Response · Authors · 2023-11-18
> **Reply to Reviewer PfVM (3/4)**
>
> > Q4. "How does the backward reasoning compare to verification of answers? E.g. Once you generate multiple candidate answers using forward reasoning, you can ask a model to “verify” the candidate answer given the whole context (Similar to verifier for GSM8K in Cobbe et al 2021)"
>
> **A4.** Compared with Cobbe et al. (2021) [9] which trains an LLM for verifying answers, our FOBAR has two advantages.
> (i) **(training-free)** Compared with training an LLM for verifying candidate answers [9], which is computationally expensive and labor-intensive in collecting extra annotation data, backward reasoning for verifying is training-free and requires no additional data collection.
> (ii) **(more effective)** As training the GPT-3 (175B) is extremely expensive and their code is not publicly available, we compare our FOBAR with the result reported in Figure 5 of [9], where the candidate answers are generated by GPT-3. Table below shows the accuracy of GSM8K. As shown, FOBAR consistently performs much better than the trained verifier (+14.8).
>
> \begin{array}{lc}
> \hline
> \text{Training GPT-3 (175B) for Verification [9]} & 56.0  \newline
> \text{FOBAR (text-davinci-003 + CoT)} & 70.8 \newline
> \text{FOBAR (text-davinci-003 + ComplexCoT)} & 78.7 \newline
> \text{FOBAR (GPT-3.5-Turbo + CoT)} & 85.1 \newline
> \text{FOBAR (GPT-3.5-Turbo + ComplexCoT)} & 87.4 \newline
> \text{FOBAR (GPT-4 + CoT)} & 95.4 \newline
> \text{FOBAR (GPT-4 + ComplexCoT)} & 96.4 \newline
> \hline
> \end{array}
>
> ---
>
> > Q5.Regarding my concern about OpenAI APIs getting updated regularly, have the authors used multiple different seeds for the experiments?
>
> **A5.** (i) As querying OpenAI's models is **costly**, we did not repeat the experiments using different seeds. This indeed is a **standard** practice in the CoT-based research literature (e.g., Self-Consistency [4], ComplexCoT [5], Least-to-Most Prompting [6], Tree-of-Thoughts [7]).
>
> (ii) **Additional experiment.**  To further resolve the reviewer's concern, we conducted an additional experiment on GSM8K using GPT-3.5-Turbo with ComplexCoT prompting. We repeat the experiment with three different seeds. Table below shows the testing accuracy. As can be seen, **FOBAR performs better than Self-Consistency on all three seeds**. Furthermore, **the improvement of FOBAR over Self-Consistency is statistically significant** (according to the pairwise t-test, with a p-value of 0.0013).
>
> \begin{array}{lcccc}
> \hline
> & \text{seed 1} & \text{seed 2} & \text{seed 3} & \text{mean $\pm$ std} \newline
> \hline
> \text{Self-Consistency} & 86.4 &86.7 & 86.2 & 86.43\pm 0.25 \newline
> \text{FOBAR} & 87.4 &87.6& 87.1 & 87.37\pm 0.25 \newline
> \hline
> \end{array}
>
> (iii) **The improvements of FOBAR over the baselines are large and sufficient to confirm FOBAR is a better verifier**. As shown in our Table 1, FOBAR with ComplexCoT prompting consistently achieves the highest accuracy. When using CoT as the base prompt, FOBAR always has a higher accuracy than the baselines. These consistent improvements are sufficient to validate FOBAR is more effective in verification.

---

> ### Author Response · Authors · 2023-11-18
> **Reply to Reviewer PfVM (4/4)**
>
> > Q6. "Given some known LLM behaviors and order of chains of arithmetic reasoning, do you think there will be a noticeable difference in performance if only the numbers at the end of the reasoning chains were masked to be predicted compared to numbers at the beginning of the reasoning?"
>
> **A6.** There might be some misunderstanding. We mask the number in the question (as shown in Example 3.1 in the submission), **NOT** the number in the reasoning chains. We give another example below to demonstrate how to mask numbers in the question.
>
> *Question*: Gretchen has 110 coins. There are 30 more gold coins than silver coins. How many gold coins does Gretchen have?
>
> *Backward Question*: Gretchen has **x** coins. There are 30 more gold coins than silver coins. How many gold coins does Gretchen have? If we know the answer to the above question is $\hat{A}_c$, what is the value of unknown variable **x**?
>
> *Backward Question*: Gretchen has 110 coins. There are **x** more gold coins than silver coins. How many gold coins does Gretchen have? If we know the answer to the above question is $\hat{A}_c$, what is the value of unknown variable **x**?
>
> ---
>
> ## **References**
>
> [1] Jason Wei, Xuezhi Wang, Dale Schuurmans, Maarten Bosma, Ed Chi, Brian Ichter, Fei Xia, Quoc Le, and Denny Zhou. Chain of thought prompting elicits reasoning in large language models. Neural Information Processing Systems, 2022.
>
> [2] Srivastava et al. (400+ authors). Beyond the imitation game: Quantifying and extrapolating the capabilities of language models. Transactions on Machine Learning Research, 2023.
>
> [3] Tianci Xue, Ziqi Wang, Zhenhailong Wang, Chi Han, Pengfei Yu, and Heng Ji. RCoT: Detecting and rectifying factual inconsistency in reasoning by reversing chain-of-thought. Preprint arXiv:2305.11499, 2023.
>
> [4] Yixuan Weng, Minjun Zhu, Shizhu He, Kang Liu, and Jun Zhao. Large language models are reasoners with Self-Veriﬁcation. Preprint arXiv:2212.09561, 2022.
>
>
> [5] Xuezhi Wang, Jason Wei, Dale Schuurmans, Quoc V Le, Ed H. Chi, Sharan Narang, Aakanksha Chowdhery, and Denny Zhou. Self-Consistency improves chain of thought reasoning in language models. In International Conference on Learning Representations, 2023.
>
> [6] Yao Fu, Hao Peng, Ashish Sabharwal, Peter Clark, and Tushar Khot. Complexity-based prompting for multi-step reasoning. In International Conference on Learning Representations, 2023.
>
> [7] Denny Zhou, Nathanael Scharli, Le Hou, Jason Wei, Nathan Scales, Xuezhi Wang, Dale Schuurmans, Claire Cui, Olivier Bousquet, Quoc V Le, and Ed H. Chi. Least-to-most prompting enables complex reasoning in large language models. In International Conference on Learning Representations, 2023.
>
> [8] Shunyu Yao, Dian Yu, Jeffrey Zhao, Izhak Shafran, Thomas L Grifﬁths, Yuan Cao, and Karthik Narasimhan. Tree of Thoughts: Deliberate problem solving with large language models. In Neural Information Processing Systems, 2023.
>
> [9] Karl Cobbe, Vineet Kosaraju, Mohammad Bavarian, Mark Chen, Heewoo Jun, Lukasz Kaiser, Matthias Plappert, Jerry Tworek, Jacob Hilton, Reiichiro Nakano, Hesse Christopher, and Schulman John. Training veriﬁers to solve math word problems. Preprint arXiv:2110.14168, 2021.
>
> [10] Takeshi Kojima, Shixiang Shane Gu, Machel Reid, Yutaka Matsuo, and Yusuke Iwasawa. Large language models are zero-shot reasoners. In Neural Information Processing Systems, 2022.
>
> [11] Luyu Gao, Aman Madaan, Shuyan Zhou, Uri Alon, Pengfei Liu, Yiming Yang, Jamie Callan, and Graham Neubig. PAL: Program-aided language models. In International Conference on Machine Learning, 2023.

---

> > ### Comment · Reviewer_PfVM · 2023-11-21
> > **Thanks for the responses**
> >
> > My concern is regarding the extension of the method to other tasks. You mentioned you would “mask informative word/token/character”. How straightforward and beneficial would the method be? Given your example, it seems that for each task one would have to think about what would be the most beneficial part to mask from the input where other similar verification methods don’t need this kind of manual design.
> >
> > My question about the novelty was about the verification part of the method which still seems like a special case of RCoT or similar methods which predict the question/query based on candidate answers.
> >
> > Self-Consistency [5] and ComplexCoT [6] amongst other methods experiment with different families of language models. Given the inconsistent results or small improvements from Table 1 in the submission, the submission would benefit from including other families of models (where experiments could be cheaper to carry out).
> >
> > I’m not sure if not repeating experiments with multiple seeds is a “standard” practice in LLM research [12] and [13] to name a few.
> >
> >
> > Regarding Q6, you’re right. Would there be a noticeable difference in performance if only numbers early in the question are masked compared to numbers towards the end of the questions?
> >
> >
> > [12] Alessandro Sordoni, Xingdi Yuan, Marc-Alexandre Côté, Matheus Pereira, Adam Trischler, Ziang Xiao, Arian Hosseini, Friederike Niedtner, Nicolas Le Roux. Deep Language Networks: Joint Prompt Training of Stacked LLMs using Variational Inference. Neurips 2023
> >
> > [13] Yao Lu, Max Bartolo, Alastair Moore, Sebastian Riedel, Pontus Stenetorp. Fantastically Ordered Prompts and Where to Find Them: Overcoming Few-Shot Prompt Order Sensitivity. ACL 2022

---

> > > ### Author Response · Authors · 2023-11-21
> > > **Reply to further comments (2/2)**
> > >
> > > > Q11. Regarding Q6, you’re right. Would there be a noticeable difference in performance if only numbers early in the question are masked compared to numbers towards the end of the questions?
> > >
> > > **A11.** We conducted the suggested experiments to study whether the performance of FOBAR is sensitive to the position of masked numbers, i.e., the first $N/2$ numbers (beginning), the last $N/2$ numbers (end), and all numbers. The table below shows the testing accuracy (averaged over six tasks). As can be seen, the performance of FOBAR is **not sensitive** to the position of numbers. Moreover, All three variants of FOBAR perform better than Self-Consistency.
> > >
> > >
> > > \begin{array}{l| c| c|c}
> > > \hline
> > > &
> > > \text{text-davinci-003} &
> > > \text{GPT-3.5-Turbo} & \text{GPT-4} \newline
> > > & \text{CoT}\quad         \text{ComplexCoT} &   \text{CoT}\quad         \text{ComplexCoT} & \text{CoT}\quad         \text{ComplexCoT}\newline
> > > \hline
> > > \text{Self-Consistency} & 81.4\quad   \quad\quad\quad             83.2 & 85.5\quad   \quad\quad\quad        86.0 & 91.1  \quad   \quad\quad\quad              91.6 \newline
> > > \hline
> > > \text{FOBAR (Beginning)} & 83.3     \quad   \quad\quad\quad           84.5 & 86.6  \quad   \quad\quad\quad              86.5 & 91.3   \quad   \quad\quad\quad             91.8 \newline
> > > \text{FOBAR (End)} & 82.7  \quad   \quad\quad\quad              84.5 & 86.3     \quad   \quad\quad\quad           86.5 & 91.4  \quad   \quad\quad\quad              91.7  \newline
> > > \text{FOBAR (All)} &  83.5   \quad   \quad\quad\quad             85.0 &  86.6  \quad   \quad\quad\quad              87.0 &  91.4   \quad   \quad\quad\quad             91.9 \newline
> > > \hline
> > > \end{array}

---

> ### Author Response · Authors · 2023-11-21
> **Any other concerns?**
>
> Dear Reviewer PfVM,
>
> We would like to thank you again for your detailed reviews.
> We hope that we have satisfactorily addressed your concerns.
>
> Given that your current score is 3, we would be grateful if you could let us know whether our response has addressed your concerns or if you still have any other questions.
>
> We would be happy to do any follow-up discussion or address any additional comments.
>
> Best,
>
> The authors

---

> ### Author Response · Authors · 2023-11-21
> **Reply to further comments (1/2)**
>
> Thanks for your further comments. We address your concerns as follows.
>
> > Q7. My concern is regarding the extension of the method to other tasks. You mentioned you would “mask informative word/token/character”. How straightforward and beneficial would the method be? Given your example, it seems that for each task one would have to think about what would be the most beneficial part to mask from the input where other similar verification methods don’t need this kind of manual design.
>
> **A7.** (i) Please note that the focus of this work is on **mathematical tasks, as emphasized in the title.** (ii) For mathematical tasks, the proposed template and FOBAR are **very general** (masking a number and predicting it). (iii) The two examples given in A1 are intended to demonstrate the possibility of extending our method to non-arithmetic tasks, which is **NOT the focus of this work**.
> (iv) For mathematical tasks, numbers are informative; However, for non-arithmetic tasks, how to select information words/marks/characters still needs to be manually designed, as two examples are given in A1. Extending FOBAR to non-arithmetic tasks is **future work**. Currently, our FOBAR focuses on mathematical tasks and does **NOT** try to solve **all verification problems**.
>
> ---
>
> > Q8. My question about the novelty was about the verification part of the method which still seems like a special case of RCoT or similar methods which predict the question/query based on candidate answers.
>
> **A8.** We have discussed our FOBAR with RCoT in Related Work of the submission and the above reply A2. We reiterate as follows:
>
> `Similarity`: Both RCoT and FOABR **use LLMs** to reconstruct/predict a missing condition in questions for verification.
>
> `Differences` (as summarized in the Table below):
>
> (i) **Where to mask**:
> - RCoT masks **a sequence of tokens**
> - FOBAR masks **a number** in the question.
>
> (ii) **How to check factual consistency**
> - RCoT needs to check the consistency of a sequence of tokens, which is very challenging. Hence, RCoT needs **three complex steps** (problem decomposition, condition comparison, question comparison) and queries the LLM $nm$ times, which is **expensive** ($m$ and $n$ are the lengths of conditions in the original and reconstructed questions, respectively)
> - FOBAR **simply** checks whether the masked number is predicted exactly by **string comparison**, which is **free**.
>
> (iii) **How to verify**
> - RCoT **uses backward verification alone**
> - FOBAR **combine both forward and backward reasoning** for verification.
>
>
> (iv) **Performance**
> - **Three complex** checking steps lead to **inacurate verification** in RCoT.
> - As shown in the table below (or Table 1 of the submission), FOBAR has **much higher** accuracy than RCoT, e.g., SVAMP (**+3.6**), GSM8K (**+2.8**), AQuA (**+10.1**).
>
> \begin{array}{c|c|c}
> \hline
> & \text{RCoT} & \text{FOBAR} \newline
> \hline
> \text{where to mask} & \text{a sequence of tokens} & \text{a number} \newline
> \hline \text{how to check factual consistency} & \text{three complex steps} & \text{simple string comparison} \newline
> \hline
> \text{how to verify} & \text{use backward verification alone} & \text{combine forward and backward reasoning together} \newline
> \hline
> \text{performance} & \text{inaccurate} & \text{much more accurate} \newline
> \text{Accuracy on SVAMP} & \text{84.2} & \text{88.9 (+3.6)} \newline
> \text{Accuracy on GSM8K} & \text{84.6} & \text{87.4 (+2.8)} \newline
> \text{Accuracy on AQuA} & \text{53.3} & \text{63.4 (+10.1)} \newline
> \hline
> \end{array}
>
> ---
>
> > Q9. Self-Consistency [5] and ComplexCoT [6] amongst other methods experiment with different families of language models. Given the inconsistent results or small improvements from Table 1 in the submission, the submission would benefit from including other families of models (where experiments could be cheaper to carry out).
>
> **A9.** As shown in Table 1, for all **three** LLMs, our FOBAR achieves **consistently** higher accuracy (averaged over **six** tasks) than previous SOTA. Note that the improvements are **NOT** marginal, e.g., FOBAR has large improvements on MultiArith (+4.1), SVAMP (+3.7), and GSM8K (+2.9) when using *text-davinci-003* with CoT Prompting. Experimenting with other families of models is a good suggestion. However, as experiments in the submission are **extensive** (three SOTA LLMs, six datasets, two base prompts), the **consistent** results are **sufficient** to verify that FOBAR is more effective than previous methods.
>
> ---
>
> > Q10. I’m not sure if not repeating experiments with multiple seeds is a “standard” practice in LLM research [12] and [13] to name a few.
>
> **A10.** Indeed, most CoT-based stuidies (e.g., Self-Consistency [4], ComplexCoT [5], Least-to-Most Prompting [6], Tree-of-Thoughts [7]) do not conduct repeated experiments, as querying OpenAI's API is costly. In addition, we **have provided a repeated experiment** on *GSM8K* in A5 to resolve your concern.

---

### Author Response · Authors · 2023-11-22
**General Discussions with Reviewers and ACs**

Dear Reviewers and ACs,

We sincerely thank all the reviewers and ACs for taking time on our submission. We are very grateful for their constructive and valuable comments.

We have carefully responded to every raised concern and hope that our rebuttal satisfactorily addressed them. Based on the reviewers' responses, we have summarized their remaining concerns and our reply below to make it easier for the reviewers and ACs to make their judgments.

## `Reviewer PfVM`
**Major Concerns** and our reply:
- **Extension our FOBAR to non-arithmetic reasoning tasks**: Note that the focus of our work is mathematical tasks (as highlighted in the title), or **AI4Math**, **NOT non-arithmetic reasoning tasks**. To further resolve this concern, we give two examples to show the possibility of extending FOBAR to Date Understanding and Last-Letter-Concatenation. Focusing on mathematical tasks, we think our reply has resolved this concern, as recognized by Reviewer nUu4, who raises a similar concern on extending FOBAR and satisfies our reply.
- **Novelty of our FOBAR relative to previous work RCoT** (proposed by `Tianci Xue, Ziqi Wang, Zhenhailong Wang, Chi Han, Pengfei Yu, Heng Ji`, [https://arxiv.org/abs/2305.11499](https://arxiv.org/abs/2305.11499)):
    - **Similarity**: **Both use LLMs to reconstruct/predict conditions based on candidate answers**. Indeed, we should focus on what to predict and how to evaluate/verify, which are crucial for performance.
    - **Differences** are summarized in the table below. Specifically, RCoT masks **a sequence of tokens** and needs **three complex steps** to check correctness, leading to **inaccurate** verification. In contrast, FOBAR masks **a number** and checks correctness by **simple** string comparison, achieving **much higher accuracy** than RCoT.

\begin{array}{c|c|c}
\hline
& \text{RCoT} & \text{FOBAR} \newline
\hline
\text{where to mask} & \text{a sequence of tokens} & \text{a number} \newline
\hline \text{how to check factual consistency} & \text{three complex steps} & \text{simple string comparison} \newline
\hline
\text{how to verify} & \text{use backward verification alone} & \text{combine forward and backward reasoning together} \newline
\hline
\text{performance} & \text{inaccurate} & \text{much more accurate} \newline
\text{Accuracy on SVAMP} & \text{84.2} & \text{88.9 (+3.6)} \newline
\text{Accuracy on GSM8K} & \text{84.6} & \text{87.4 (+2.8)} \newline
\text{Accuracy on AQuA} & \text{53.3} & \text{63.4 (+10.1)} \newline
\hline
\end{array}

**Minor concerns** and our reply:
- **experimenting with other families of models**: Experiments in the submission are **extensive** (**three** SOTA LLMs, **six** datasets, **two** base prompts), and the **consistent** improvement of FOBAR over baselines are sufficient to verify that FOBAR is more effective.
- **repeating experiments**: As querying OpenAI's API is costly, CoT-based works rarely conduct repeating experiments, e.g., (e.g., Self-Consistency [4], ComplexCoT [5], Least-to-Most Prompting [6], Tree-of-Thoughts [7]). To resolve your concern, we **have provided a repeating experiment** on GSM8K in A5.

---

## `Reviewer YtZF`
**Major concerns** and our reply:
- **Complex cases (multiple correct values for ${\bf x}$)**: FOBAR uses backward reasoning to **facilitate** the verification of candidate answers generated by forward reasoning. By our design (step 11 in Algorithm 1), a backward reasoning chain is correct only when its prediction ${\bf x}$ is exactly the same as **the masked number** (by **string comparison**). In backward reasoning, we select the candidate answer that leads to the most correct backward reasoning chains. When there are multiple correct values for ${\bf x}$, the proposed backward reasoning can work as long as the correct candidate answer is easier for the LLM to exactly predict ${\bf x}$ as the masked number than the wrong candidate answers.
- **misunderstanding**: Accuracy of backward reasoning is **the accuracy of using backward reasoning in verifying candidate answers generated by Self-Consistency**, **NOT** the accuracy of the predicted ${\bf x}$ being a correct value (may differ from the masked number in complex cases). See A5-A6 in our reply for detailed explanation.

---

Please let us know if you have any remaining questions; we are more than happy to address them.

Thanks again for all the effort and time.

Best,

Authors

---

### Meta-Review · Area_Chair_gq7V · 2023-12-05

**Metareview:**

This paper presents a method for mathematical reasoning tasks that integrates both forward and backward reasoning. After an LM predicts an answer, the LM is used to regenerate masked values from the question, thereby trying to do round-trip confirmation of the computation.  Compared to past work like RCoT, this is different in that only these specific masked values are regenerated.  The method gives consistent gains over self-consistency (frequently the strongest baseline), although these gains are lower on stronger models.

The idea in this paper is quite nice. nUu4 articulates this point well, that backward reasoning is a nice conceptual idea that is likely to have impact going forward. There is a big empirical improvement over RCoT and the method is conceptually differentiated (argued by a reviewer in private discussion).

One of the main critiques of this paper is the narrow focus on mathematical reasoning. The authors give some examples of other tasks in the rebuttal, but these are still very "computational" tasks, just involving manipulation of symbols other than mathematical ones. I am a bit torn on this critique, but I do think it's relevant when comparing to self-consistency, which relies on any notion of equality and not a particular masking strategy.

Ultimately, the reviewers remained divided on this paper even after some discussion. The decision came down to the fact that the method is somewhat specialized and the improvements over self-consistency are small. While the concept of the paper is good, the package of (concept, novelty, results) doesn't quite rise to the bar of ICLR.

**Justification For Why Not Higher Score:**

See comments on reviewer discussion above. It could be accepted, but the novelty+results+excitement of the method aren't quite there.

**Justification For Why Not Lower Score:**

N/A

---

### Decision · Program_Chairs · 2024-01-16

Reject